# Genetic coupling of enhancer activity and connectivity in gene expression control

Helen Ray-Jones [1,2,3,4,14] ✉, Chak Kei Sung[1,2,15], Lai Ting Chan [3,4],
Alexander Haglund [5], Pavel Artemov[1,2], Monica Della Rosa[1,2,16], Luminita Ruje[1,2],
Frances Burden[6,7,17], Roman Kreuzhuber [6,7,8,18], Anna Litovskikh [1,2,19],
Eline Weyenbergh[3,4,20], Zoï Brusselaers[3,4,21], Vanessa Xue Hui Tan[1,2,22],
Mattia Frontini [6,7,9], Chris Wallace [10,11], Valeriya Malysheva[1,2,3,4,23,24],
Leonardo Bottolo [11,12,13,23,24] ✉, Elena Vigorito [11,23,24] &
Mikhail Spivakov [1,2,24] ✉

Gene enhancers often form long-range contacts with promoters, but it remains unclear if the activity of enhancers and their chromosomal contacts are mediated by the same DNA sequences and recruited factors. Here, we study the effects of expression quantitative trait loci (eQTLs) on enhancer activity and promoter contacts in primary monocytes isolated from 34 male individuals. Using eQTL-Capture Hi-C and a Bayesian approach considering both intra- and inter-individual variation, we initially detect 19 eQTLs associated with enhancer-eGene promoter contacts, most of which also associate with enhancer accessibility and activity. Capitalising on these shared effects, we devise a multi-modality Bayesian strategy, identifying 629 "trimodal QTLs" jointly associated with enhancer accessibility, eGene promoter contact, and gene expression. Causal mediation analysis and CRISPR interference reveal causal relationships between these three modalities. Many detected QTLs overlap disease susceptibility loci and influence the predicted binding of myeloid transcription factors, including SPI1, GABPB and STAT3. Additionally, a variant associated with *PCK2* promoter contact directly disrupts a CTCF binding motif and impacts promoter insulation from downstream enhancers. Jointly, our findings suggest an inherent genetic coupling of enhancer activity and connectivity in gene expression control relevant to human disease and highlight the regulatory role of genetically determined chromatin boundaries.

Distal DNA regulatory elements such as transcriptional enhancers play crucial roles in spatiotemporal gene expression control and are enriched for genetic variants that associate with molecular and cellular traits, such as the risk of common diseases (typically ascertained by genome-wide association studies [GWAS])[1,2] and the expression levels of specific genes ('expression quantitative trait loci', eQTLs)[3–7]. The binding of transcription factors (TFs) to these elements in response to intrinsic or extrinsic cues promotes a cascade of events that include

the recruitment of chromatin remodelers and histone modifiers[8,9]. Jointly, these events result in the establishment of active chromatin signatures and the engagement of RNA polymerase at these loci. Consequently, the active state of an enhancer is transferred to the promoters of one or more target genes, leading to their increased transcription[10–12].

The transfer of activating signals from distal enhancers to promoters typically (though not always) depends on connections between

these elements within the nucleus. These connections can be established through direct 3D chromosomal contacts or potentially through other mechanisms, such as phase separation[12–15]. Previous studies that compared the dynamics of enhancer activation, enhancer-promoter connections and gene expression across cell types and conditions have found that these events are often correlated, although cases whereby enhancer-promoter contacts precede enhancer activation and gene induction are also common[5,13,16–18]. The precise molecular mechanisms underpinning the observed relationships between these phenomena, however, remain incompletely understood.

The canonical mechanism for generating contacts between distal DNA loci relies on the architectural proteins cohesin and CTCF[19,20]. Cohesin creates and continuously extrudes chromatin loops, while CTCF binds to DNA in a sequence-specific manner and constrains loop extrusion. Jointly, these phenomena promote the formation of topologically associated domains (TAD) that largely insulate enhancer-promoter contacts[21,22]. However, the interaction between cohesin and CTCF cannot fully explain the phenomenon of enhancer-promoter communication, as many enhancers and promoters localise away from TAD boundaries, often lack canonical CTCF binding sites and do not fully dissolve after cohesin or CTCF degradation[23–26]. In addition, various transcription factors involved in enhancer and gene activation were shown to affect enhancer-promoter contacts without engaging directly with these architectural proteins[25,27–30]. These observations raise the possibility that the ability of enhancers to form long-range promoter contacts (their 'connectivity') is inherently linked with enhancer activation at the genetic level, rather than being encoded and facilitated separately.

To address this possibility, we focused on enhancers that harbour eQTL variants associated with distal gene expression in monocytes—an abundant and readily available innate immune cell type with a high clinical significance, for which high-confidence eQTL meta-analysis data are available[31]. We asked whether the known transcriptional effects of these variants coincided with their shared or distinct impact on enhancer activity and enhancer-promoter contacts. We performed ATAC-seq and high-resolution Capture Hi-C targeted at eQTL-harbouring regions in monocytes isolated from multiple healthy donors and adapted Bayesian methods to identify genetic associations with molecular phenotypes at increased power. We detected abundant shared effects of enhancer variants on promoter contacts, local chromatin accessibility (as a proxy of activity) and target gene expression. We then obtained molecular and in silico evidence for causal relationships between these modalities using causal mediation analysis and CRISPR-mediated perturbation. Variants with shared effects on enhancer accessibility, connectivity and gene expression localised away from CTCF binding sites, often disrupted the predicted binding of a diverse range of myeloid transcription factors and overlapped disease susceptibility loci. In addition, we identified a distal eQTL variant with opposing effects on promoter contact and gene expression and no detectable impact on chromatin accessibility. We show that this variant disrupts the canonical CTCF binding motif and likely acts by perturbing the insulation of its target gene promoter from upstream distal enhancers. Jointly, our findings suggest an inherent genetic coupling of enhancer activity and connectivity in gene expression control with relevance to human disease and highlight the regulatory role of genetically determined chromatin boundaries.

## Results

### A compendium of eQTL-anchored chromosomal contacts, open chromatin and gene expression in human primary monocytes isolated from multiple individuals

To profile eQTL-anchored chromatin contacts at high throughput and resolution, we employed Capture Hi-C (CHi-C) using a frequently-cutting DpnII restriction enzyme. We designed a custom capture system targeting 1197 distal eQTLs in unstimulated monocytes (located at least 10 kb away from their target gene promoters) that were detected in a multi-cohort monocyte eQTL study in 1480 individuals[31,32] (Fig. 1A and Supplementary Data 1). Our capture system targeted the lead eQTLs, as well as the variants in tight linkage disequilibrium (LD) with them ($r^2 \geq 0.9$) that mapped within known regulatory regions (see 'Methods' for details). For each eQTL, we aimed to capture the promoter(s) of the associated gene ('eGene') and those of distance-matched 'non-eGenes', whose expression was not associated with the eQTL. We successfully designed capture probes for 1074 eGenes and 4718 control genes, respectively (Fig. 1B; see 'Methods' for details). The distances between the contact eQTLs and the respective eGene promoters ranged from ~32.5 to ~283 kb, with a median of 72.3 kb. Following sequencing and quality control, the eQTL CHi-C experiment generated a minimum of 9.5 million unique valid on-target reads per donor (median 11.5 million, Supplementary Data 2). By combining the CHi-C data from all 34 donors, we initially obtained a high-coverage consensus dataset consisting of just over 470 million on-target unique reads. Using the CHiCAGO algorithm[33,34], we identified 642,552 significant (score ≥ 5) contacts at DpnII-fragment resolution and 398,464 contacts in bins of 5 kb in this consensus dataset.

To profile chromatin accessibility, we adapted ATAC-seq for crosslinked input material (see 'Methods') and performed the assay on the same formaldehyde-fixed samples as used for CHi-C. The ATAC-seq datasets generated on crosslinked chromatin were of high quality and had a high degree of overlap with two published datasets in primary monocytes[35,36] (Supplementary Fig. 1 and Supplementary Data 3). Using a Hidden Markov Model-based pipeline for ATAC-seq[37], we detected 74,373 open chromatin peaks in the ATAC-seq consensus data pooled across individuals. Finally, we profiled gene expression using RNA-seq and genotyped each donor (see Supplementary Data 4).

Using appropriate metrics separately for each assay (CHi-C, ATAC-seq and RNA-seq), we confirmed that overall, the generated data were highly concordant across the cohort (Supplementary Fig. 2A–C). Next, integrating the three assays, we found that, as expected, the regions contacted by the captured fragments (containing either eQTL sites or gene promoters) were enriched for open chromatin peaks, as well as for active histone marks detected in other studies in monocytes[38] (Supplementary Fig. 2D). We also observed a positive correlation between the level of gene expression and the number of active regions contacted by a gene's promoter (Spearman's rho = 0.22, p value < $2.2 \times 10^{-16}$), consistent with previous reports[5,39] (Fig. 1C). Jointly, these consensus data provide a high-coverage compendium of chromosomal contacts, chromatin accessibility and gene expression in human primary monocytes.

### Distal eQTLs engage in chromosomal contacts with their target genes

We next asked whether eQTLs preferentially shared topologically associating domains (TADs) and formed chromosomal contacts with their target eGenes compared with distance-matched control genes. We called TADs using Hi-C data from two donors in the cohort (Supplementary Data 5; see Supplementary Data 2 for donor information). Overall, 95% of eQTLs and their respective eGene promoters shared the same TAD, while a significantly lower proportion of eQTLs shared the same TAD with the promoters of distance-matched control genes (85%, exact Fisher test p value = $3.6 \times 10^{-13}$; Fig. 1D). Moreover, even within the same TAD, eQTL contacts with eGene promoters tended to have higher CHiCAGO interaction scores than those with control gene promoters (Wilcoxon paired test p value < $2.2 \times 10^{-16}$; Fig. 1E), consistent with previous findings[5,40]. At a CHiCAGO score cutoff of 5, around 70% of eQTLs (816/1197) had significant chromosomal contacts with the respective eGene promoter, detected either in at least one individual or in the high-coverage consensus CHi-C dataset (see Fig. 1F–H for examples). As a complementary approach for detecting functional enhancer-

promoter contacts, we additionally used the threshold-free 'Activity-By-Contact' (ABC) method[41,42], which we recently adapted to CHi-C data (CHi-C ABC)[39]. In this approach, the effects of an enhancer on a target promoter are estimated as the product of local enhancer activity and enhancer-promoter contact frequency. Using ATAC-seq data as a proxy of enhancer activity, ABC predicted functional links between 14,413 putative enhancers and 4440 genes that were captured in the

CHi-C experiment (at an ABC score threshold of 0.012 selected as described in the 'Methods'; Supplementary Data 6). However, the ABC-detected pairings only connected 175/1197 (14.6%) of our targeted eQTLs with their respective eGenes, just 21 of which were not identified by CHiCAGO. The small number of eQTL-promoter contacts detected by ABC is likely due to our focus on distal eQTLs (median eQTL−eGene TSS distance: 57 kb), while ABC preferentially identified short-range

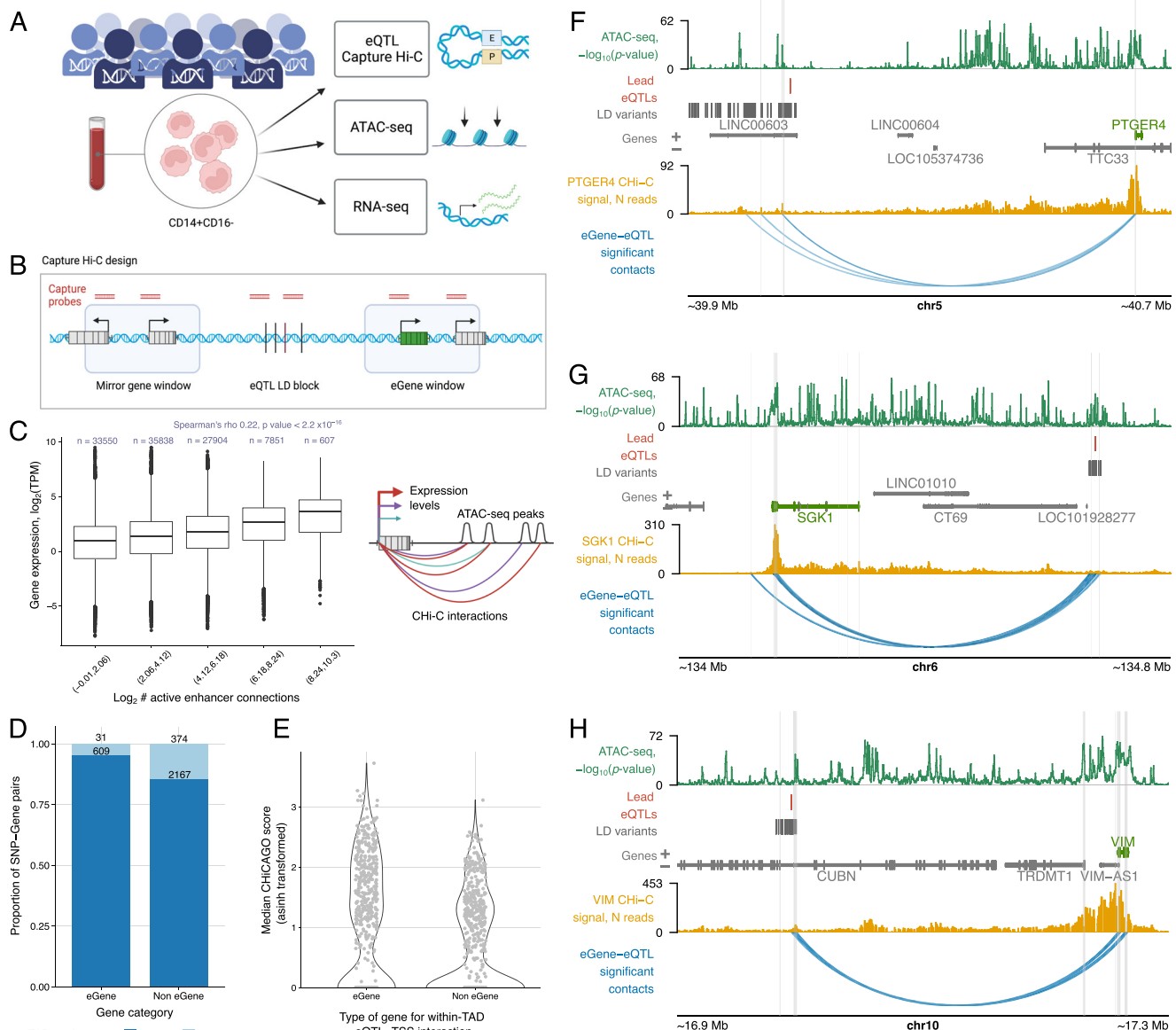

**Fig. 1 | A compendium of eQTL CHi-C contacts and accessibility in primary monocytes. A** Overview of the main data collection steps. Created in BioRender: https://BioRender.com/v74s856. **B** Design of the eQTL CHi-C experiment. We designed capture probes targeting *DpnII* fragments harbouring previously identified lead eQTLs in monocytes. We also included variants in tight LD with the lead eQTLs in regulatory regions, eGene promoters and the promoters of distance-matched 'non-eGenes', which were similar distances from the eQTLs but not associated with their expression. Created in BioRender: https://BioRender.com/z90t537. **C** Relationship between the number of interacting enhancers and gene expression. Two-sided Spearman's rank correlation was performed on log₂(number active enhancers) against log₂(expression TPM) for 5729 genes in 34 samples. Boxplots show 25th, 50th and 75th percentiles, with upper and lower whiskers to the largest or smallest value no further than 1.5 x the interquartile range from the hinge. Graphic created in BioRender: https://BioRender.com/s35y475. **D** Degree of TAD sharing between eQTLs and eGenes or eQTLs and non-eGenes.

**E** Inverse hyperbolic sine (asinh)-transformed median CHiCAGO scores for interactions between eQTLs and eGenes or non-eGenes within the same TAD. The score for the eGene is shown against the median score for all captured control genes, per eQTL, including cases where the score was zero. Examples of eQTLs intersecting ATAC-seq peaks and interacting with the eGenes: *PTGER4* (**F**), *SGK1* (**G**) and *VIM* (**H**). ATAC data are presented as −log₁₀(p value) for the consensus dataset (detected by Genrich[140]). eGenes are highlighted in green. Contact profiles show the number of reads for each other-end fragment contacting the fragment containing the eGene promoter in the consensus dataset. The eQTL-eGene significant contacts were called using the shown consensus CHi-C interactions (CHiCAGO statistical algorithm on consensus data (score ≥ 5), at *DpnII*-fragment level). Interactions are restricted to those involving the eQTL or a SNP in tight LD and the eGene promoter. Baited regions are shown as a grey highlight. **F**−**H** were plotted using the Plotgardener R package[125]. Source data for **C**−**E** are available on OSF[168].

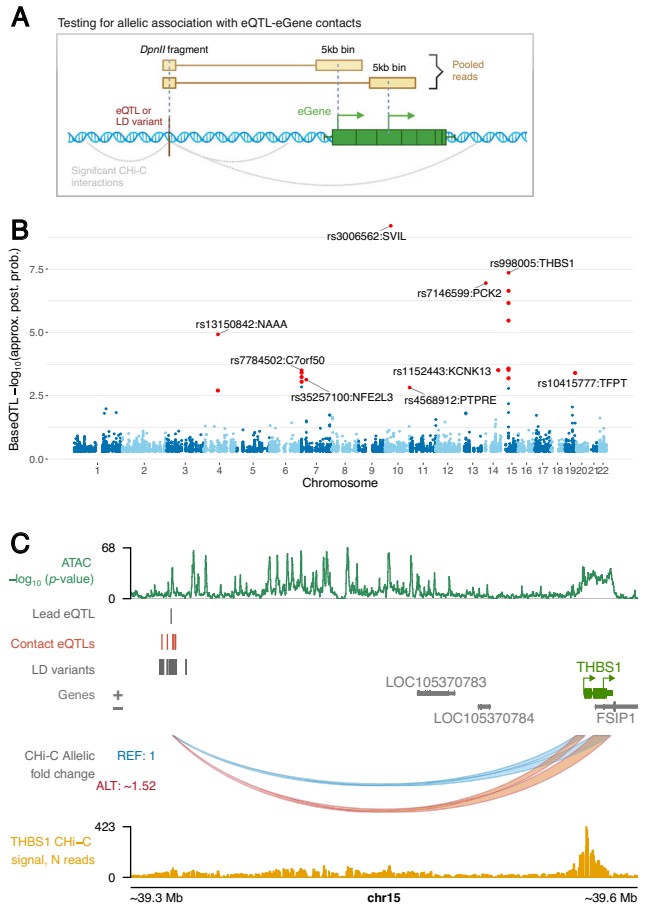

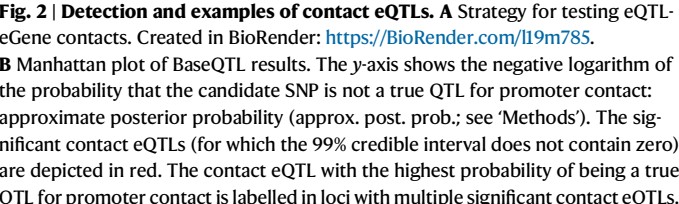

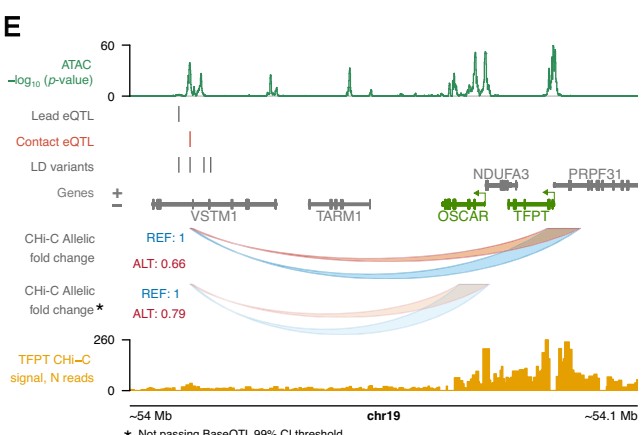

**Fig. 2 | Detection and examples of contact eQTLs. A** Strategy for testing eQTL-eGene contacts. Created in BioRender: https://BioRender.com/l19m785. **B** Manhattan plot of BaseQTL results. The *y*-axis shows the negative logarithm of the probability that the candidate SNP is not a true QTL for promoter contact: approximate posterior probability (approx. post. prob.; see 'Methods'). The significant contact eQTLs (for which the 99% credible interval does not contain zero) are depicted in red. The contact eQTL with the highest probability of being a true QTL for promoter contact is labelled in loci with multiple significant contact eQTLs.

**C–E** Genomic visualisation of significant contact eQTLs in the *THBS1*, *NAAA* and *TFPT* loci. The arches, whose heights correspond to the allelic fold change in contact, show the tested contacts between the eQTL and promoter(s) of the eGene (one eQTL-containing *DpnII* bait fragment is shown in each case). CHi-C read counts at *DpnII* fragment resolution are shown from the viewpoint of the eGene promoter (yellow signal tracks). ATAC data is presented as −log₁₀(*p* value) on consensus data (detected by Genrich[140]). **C–E** were plotted using the Plotgardener package[125]. CI, confidence interval.

promoter contacts (median distance ~23 kb), consistent with previous results obtained with this method in other systems[39,41,42].

Taken together, these findings indicate that eQTLs commonly and preferentially contact the promoters of their target eGenes, suggesting that chromosomal contacts may be involved in mediating the genetic effects of these loci on eGene expression.

### Association of expression QTLs with differential promoter contacts

To detect the potential genetic effects of eQTLs on their chromosomal contacts with the eGene promoters from our eQTL CHi-C data, we adapted an allele-specific Bayesian QTL mapping methodology, BaseQTL, that considers both within-individual and between-individual allelic effects and explicitly models mappability bias and was shown to have a lower error rate and increased power compared with alternative methods[43].

BaseQTL was originally developed for eQTL calling[43]. Here, we extended the BaseQTL model for calling QTLs based on ATAC-seq and CHi-C read counts (Fig. 2A), which included directly accounting for the uncertainty of the QTL genotypes called by imputation (see 'Methods'). The density of CHi-C read counts across individuals allowed us to

query genetic associations at 1110 out of the 1197 targeted eQTL loci, including 974 lead eQTLs and 8759 proxy variants in tight LD (*r²* ≥ 0.9).

At a stringent credible interval (CI) threshold of 99%, we detected 19 significant variants in 9 independent loci associated with differential contacts between 13 eQTL-containing *DpnII* fragments and the promoters of 9 eGenes. We term these variants 'contact eQTLs' (Fig. 2B, Supplementary Fig. 3 and Supplementary Data 7).

The thrombospondin 1 (*THBS1*) locus had the largest number of contact eQTLs (seven, in tight LD with each other [*r²* ≥ 0.9]). The alternative alleles of these variants were associated with stronger contacts with *THBS1* promoters located ~250 kb away from them (Fig. 2C). We validated the effect of one of these variants, rs2033937, on chromosomal contacts with *THBS1* using allele-specific 4C-seq in a subset of heterozygous samples (Supplementary Fig. 4A). In the *C7orf50* and *NAAA* loci, we also detected multiple contact eQTLs in tight LD with each other, whilst only one contact eQTL was detected within each of the remaining six loci (*NFE2L3*, *PTPRE*, *SVIL*, *KCNK13*, *PCK2* and *TFPT*) (*NAAA* locus illustrated in Fig. 2D). Notably, the *TFPT* contact eQTL (rs10415777) was also in tight LD with an eQTL (rs35154518) for another gene, *OSCAR*. The expression levels of *TFPT* and *OSCAR* were correlated across the donors in our cohort (Pearson's

correlation $r = 0.54$, $p$ value $= 0.001$), suggesting a shared genetic control of these two genes. Moreover, rs10415777 showed the same direction of effect for contacts with both *TFPT* and *OSCAR* (Fig. 2E). Although the effect of rs10415777 on *OSCAR* did not pass our stringent significance threshold, these observations raise the possibility that rs10415777 may affect the promoter contacts of the regulatory region encompassing this variant with the promoters of both genes.

We asked whether eQTLs were more likely to influence promoter contacts compared with variants that contacted promoters in 3D, but did not associate with gene expression. To test this, we considered a set of 11,582 control (non-eQTL) variants with significant evidence of promoter contact in our consensus CHi-C dataset (CHiCAGO score ≥5) that are not known to be eQTLs in monocytes[32] or across blood cells[31] and assessed their association with the promoter contact strength using BaseQTL (Supplementary Data 7C). Despite the 1.4-fold larger number of non-eQTL variants compared with that of the eQTLs included in the analysis, only two non-eQTLs, mapping to a single genomic locus, showed association with contact strength (rs7905438 and rs7917620 contacting the promoter of *RNU6-6P* from a distance of 1 kb). We confirmed the significant enrichment of genetic effects on promoter contact among eQTLs compared with promoter-interacting non-eQTLs using a permutation test that compared the respective posterior probabilities of association, accounting for read coverage (one-sided $p$ value $< 1 \times 10^{-4}$; see 'Methods').

## Most contact eQTLs associate with local chromatin activity

To probe the relationship between the genetic effects of enhancer connectivity and activity on gene expression, we additionally detected QTLs associated with differential ATAC-seq signals in the same individuals (ATAC-QTLs; Fig. 3A and Supplementary Data 8), as well as validated the effects of multi-cohort eQTLs using the RNA-seq data from our cohort (Supplementary Fig. 4B, C and Supplementary Data 9; see 'Methods' for details). Genome-wide, we detected 34,900 ATAC-QTL variants for 6855 ATAC-seq peaks located within a 5 kb vicinity from the peak. ATAC-QTLs directly overlapped 104/1197 (~9%) lead eQTLs profiled in the eQTL-CHiC experiment and were in tight LD ($r^2 \geq 0.9$) with an additional 119 lead eQTLs. ATAC-QTLs also directly overlapped 10/19 contact eQTLs (14 accounting for LD). Notably, the allelic effects of these variants on both contact strength and enhancer accessibility were exclusively in the same direction (Fig. 3B). In all but one locus, these effects were also in the same direction as gene expression (Fig. 3C).

To further assess the genetic effects of contact eQTLs on enhancer activity, we integrated QTL data for histone modifications in human monocytes from the Blueprint WP10 Phase II release[44]. We found that 53% (10/19) of the contact eQTLs overlapped QTLs for the nearby active histone marks H3K27ac and H3K4me1 (located within 5 kb of the contact eQTL, accounting for LD [$r^2 \geq 0.9$]; Fig. 3C). The direction of effect for the histone mark QTLs was always consistent with that of contact, accessibility and eGene expression (Fig. 3C). For instance, all contact eQTLs in the *THBS1* locus were also QTLs for one or more local features of activity (ATAC-seq, H3K27ac and H3K4me1), and the signals for all these features were stronger in the alternative allele (Fig. 3D). We observed a similar pattern of stronger enhancer activity signals associated with the alternative allele of the single contact eQTL in the *SVIL* locus (Fig. 3E). In addition to their effects on local enhancer activity, we also found that some of the contact eQTLs were associated with increased chromatin accessibility and active histone marks at the promoters of their respective eGenes. For example, contact eQTLs for *NAAA* and *KCNK13* were associated with H3K27ac levels both proximally to these variants and at the promoters of these genes (Fig. 3E, F, respectively).

In conclusion, contact eQTLs typically, but not always, associate with differential chromatin activity *in cis*, suggesting abundant shared genetic effects on enhancer activity, connectivity and gene expression.

## Joint detection of shared genetic effects on enhancer accessibility, connectivity and gene expression

The approach described above identified genetic associations with enhancer-promoter contact and showed overlap with genetic effects on other molecular phenotypes ('modalities'), despite testing each one separately. Given this overlap and our relatively small sample size, we sought to increase the power of association detection by jointly modelling all three modalities (gene expression, chromatin accessibility and eQTL-promoter contact). To detect such 'trimodal QTL' variants, we adapted a Bayesian QTL mapping framework, GUESS[45,46], developed to enhance the power to detect joint QTLs by leveraging the information contained in multiple molecular traits. Specifically, we defined 5564 windows containing 24,485 genotyped variants within ~5 kb proximity to the tested distal eQTL-eGene contacts and ATAC-seq peaks (Fig. 4A). Within each window, we searched for models consisting of one or a combination of genetic variants that jointly explained the three modalities, accounting for potential confounders (see 'Methods'). At a 5% FDR[47] in each window and following post-hoc filtering (see 'Methods'), we identified 919 partially overlapping windows containing 629 significant trimodal QTLs (Supplementary Data 10). In total, trimodal QTLs were associated with 705 eQTL-eGene contacts, 299 open chromatin peaks and 217 eGenes (Fig. 4B). The majority of windows showing significant associations (869/919, 95%) involved a single trimodal QTL ('single-QTL window'), but in some loci, two or three trimodal QTLs best explained the observed genetic effects on chromatin accessibility, contact and gene expression at a 5% FDR (Fig. 4C). The detected associations were highly enriched for the same direction of genetic effect across all three modalities, with 46% of windows exhibiting such concordant effects, compared with the 25% expected by chance for three independent modalities (binomial test, $p$ value $= 1.2 \times 10^{-74}$). In addition, over 71% of trimodal QTLs had concordant genetic effects between at least one tested ATAC-seq peak and eGene promoter contact (Supplementary Data 10). Notably, loci showing concordant effects tended to have stronger effect magnitudes (Supplementary Fig. 5A, Wilcoxon test $p$ value $= 9.7 \times 10^{-09}$).

The GUESS approach replicated the effects of 84% (16/19) of the CHi-C BaseQTL contact eQTLs on the contacts with six respective eGene promoters (*THBS1*, *NAAA*, *SVIL* and *KCNK13*, *TFPT* and *C7orf50*), either via direct variant overlap ($N = 7$) or by overlap with other variants within the considered GUESS window and in tight LD ($r^2 > 0.9$) with the lead trimodal QTL ($N = 9$). Overall, the estimated variant effects for each of the three modalities (contact, accessibility and expression) were highly correlated between the BaseQTL and GUESS approaches, while the significance of the effects was higher for GUESS, indicative of the increased power of this approach (Supplementary Fig. 4). Two examples of trimodal QTLs and their joint effects on the three modalities are shown at the Toll-like receptor 5 (*TLR5*) and Abhydrolase domain-containing protein 2 (*ABHD2*) loci (Fig. 4D, E). In each of these two cases, the best variant selected by GUESS was in tight LD with the lead eQTL from the original monocyte study ($r^2 > 0.99$).

In summary, by jointly analysing functionally related molecular traits, we have identified a large set of monocyte QTLs with shared effects on chromatin accessibility, enhancer-promoter communication and gene expression, which we release as a resource in Supplementary Data 10.

## Mediation analysis and CRISPR interference reveal causal relationships between enhancer activity, connectivity and gene expression at trimodal QTLs

The shared associations of trimodal QTLs with enhancer activity (proxied by accessibility), enhancer-promoter contacts and gene expression can reflect either independent effects of the genotype on each of the three modalities or a hierarchically causal relationship where the genotype affects one modality, which in turn mediates effects on the others.

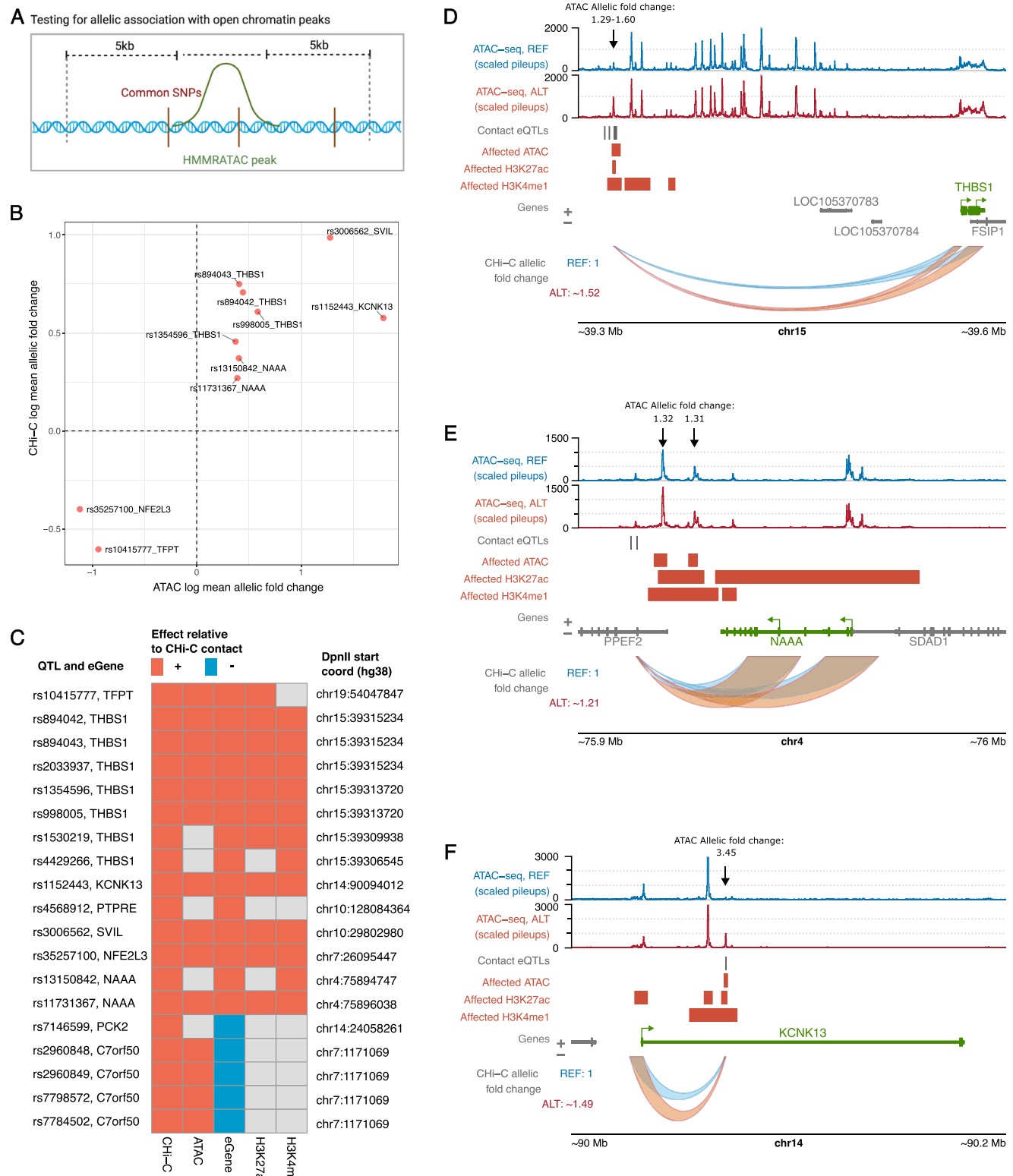

To explore causal relationships between modalities, we employed a statistical approach known as causal mediation. This approach identifies whether an independent variable (in this case, genotype) affects the outcome through an intermediate variable (mediator), accounting for sensitivity to assumption violation, partial mediation effects and potential confounders (Fig. 5A, see 'Methods' for details). We used this framework to test three non-mutually exclusive models of causal mediation at each trimodal QTL. In models I and II, chromatin

accessibility (proxied by ATAC-seq signal) was the mediator of genotype effect on (I) chromosomal contact with the eGene promoter (proxied by CHi-C signal) (Fig. 5B) or (II) eGene expression (proxied by RNA-seq signal) (Fig. 5C). In model III, chromosomal contact was the mediator of the genotype effect on gene expression (Fig. 5D). For each locus and model, we estimated the average causal mediation effect (ACME) across individuals, the average direct (non-mediated) effect (ADE) across individuals as well as the total effect (TE = ACME + ADE;

**Fig. 3 | Shared genetic effects on enhancer activity, enhancer-promoter contact and gene expression. A** Strategy for detecting ATAC QTLs with BaseQTL. Created in BioRender: https://BioRender.com/v31q004. **B** Shared allelic effects of QTLs on accessibility and promoter contact at 99% credible interval. **C** Heatmap of cross-trait effects for contact eQTLs. The positive effect allele (either REF or ALT) is shown for CHi-C, and the direction of effects in other traits (whether log allelic fold change or beta) is shown relative to this. We only show effects for features within 5 kb of the contact eQTL, and we accounted for LD ($r^2 \geq 0.9$). The ATAC QTL effects were taken from BaseQTL results within our 34-donor cohort, whereas the remainder of effects (gene expression, H3K27ac and H3K4me1) were curated from outside of our cohort. The eQTL effects were taken from the original monocyte study or from Blueprint. The histone modifications H3K27ac and H3K4me1 were taken from Blueprint WP10 Phase 2. Epigenetic mechanisms within example contact eQTL loci: (**D**) *THBS1*, (**E**) *NAAA*, **F** *KCNK13*. The red regions show upregulated peaks of ATAC, H3K27ac and H3K4me1 associated with the contact eQTL (or SNPs in LD, $r^2 \geq 0.9$); the peaks shown in these plots were not restricted to 5 kb from the contact eQTL. ATAC-seq tracks show the pileups for merged homozygous reference (blue) or homozygous alternative (red) donors for one of the contact eQTLs as an approximation of the allele-specific signal across the locus. Black arrows indicate the ATAC-seq peaks and the fold changes associated with the alternative genotype of each contact eQTL in the BaseQTL analysis (see also Supplementary Data 8). **D**–**F** were plotted using the Plotgardener package[125]. Source data for (**B**) and (**C**) are available on OSF[168].

Fig. 5A). Significant ACME and TE (ACME *p* value ≤ 0.05 and TE *p* value ≤ 0.05) were indicative of mediation. A lack of evidence for direct effects (ADE *p* value > 0.05) in combination with significant ACME and TE suggested that the mediator fully explained the relationship between the genotype and the outcome ('full mediation'). In contrast, when both ADE and ACME were significant (*p* value ≤ 0.05), this indicated that the mediator partially explained the relationship between the genotype and the outcome ('partial mediation').

Approximately ~9% (79/919) of the partially-overlapping GUESS QTL windows showed evidence for either partial or full causal mediation effects in at least one of the three models, implicating 62 trimodal QTLs and 33 eGenes in total (Supplementary Fig. 6A and Supplementary Data 11). For example, in the *THBS1* locus, the accessibility of the QTL region fully mediated the relationship between the genotype and the region's contact with the *THBS1* promoter (Model I; TE *p* value < $1 \times 10^{-16}$, ACME *p* value = 0.004 and ADE *p* value = 0.30) (Fig. 5E, F; sensitivity shown in Supplementary Fig. 6B). We confirmed this relationship experimentally by targeting CRISPR interference (CRISPRi) in a monocytic cell line to the trimodal QTL region in the *THBS1* locus. This perturbation resulted in an approximately 25% reduction in the ATAC-seq signal at the trimodal QTL region (top left plot in Fig. 5G) that corresponded with a ~22% reduction at the promoter (top right plot in Fig. 5G), consistent with the reduced activity of respective enhancers, induced by the enhancer CRISPRi perturbation. Concurrently, the chromosomal contacts of the QTL region shifted upstream of this region and away from the *THBS1* promoter (see 4C-seq locus plot in Fig. 5G and left-hand graph in Fig. 5H), mirroring the effects observed between the alleles of this QTL in primary monocytes (right-hand graph in Fig. 5H). Furthermore, these effects resulted in an approximately 30% reduction in *THBS1* expression (two-sided *T*-test, *p* value = 0.03; Fig. 5I and Supplementary Fig. 6C)

In addition, we identified a large number of cases (69 windows, implicating 52 trimodal QTLs and 23 genes, Supplementary Data 11) where chromatin accessibility partially or fully mediated the effect of genotype on gene expression (Model II). As an example, chromatin accessibility of an ATAC-seq peak at the QTL region fully mediated the relationship between the genotype and the expression of the eGene *NFE2L3* (TE *p* value < $10^{-16}$, ACME *p* value = 0.024 and ADE *p* value = 0.16) (Fig. 5J, K; sensitivity plot shown in Supplementary Fig. 6D). Finally, in a small number of cases (4 windows, implicating four trimodal QTLs and four genes, Supplementary Data 11) chromatin contacts were found to partially or fully mediate the relationship between the genotype and gene expression (model III). In the fully mediated case, a ~20 kb chromatin contact between trimodal QTLs and *SPSB1* mediated the QTL's genetic effect on *SPSB1* expression (TE *p* value = 0.012, ACME *p* value = 0.012 and ADE *p* value = 0.164) (Fig. 5L, M; sensitivity plots shown in Supplementary Fig. 6E).

Jointly, these analyses highlight hierarchical causal relationships between enhancer activity, enhancer-promoter contact and gene expression that mediate the effects of sequence variation at trimodal QTLs.

## Genetic effects on enhancer-promoter contacts are likely mediated by a diverse range of transcription factors

To determine the mechanisms underlying the phenotypic effects of the identified QTLs, we searched for evidence of protein binding at these loci. For this analysis, we combined contact eQTLs identified by BaseQTL and trimodal QTLs identified by GUESS, obtaining 641 distinct variants, to which we will refer collectively as 'contact QTLs' (cQTLs).

We first compiled evidence of protein binding from the ReMap[48] catalogue, a database of chromatin immunoprecipitation (ChIP)-seq peaks for 1171 TFs in 726 human cell types. We extracted and merged the peaks found across monocyte cell types per TF (see 'Methods'), resulting in a monocyte-specific peak set of 14 TFs. To expand the TF repertoire, we additionally generated context-aware binding predictions for 710 TFs within our monocyte ATAC-seq data using TOBIAS footprint analysis[49] and the MaxATAC deep learning framework[50], making a total peak set for 716 TFs in monocytes. Compared with all 25,151 eQTL variants considered in our study, cQTLs were significantly enriched for the ChIP-seq binding sites of 6 TFs, including key TFs involved in monocyte differentiation and activation such as SPI1 (also known as PU.1), CEBPB and STAT3, a metabolic regulator SREBP2, a Mediator-associated cell-cycle protein CDK8 and the CREBBP transcriptional coactivator (Fig. 6A, green triangles, and Supplementary Data 12). Considering the ATAC-seq predicted peaks, an additional 37 TFs were enriched at cQTLs (Fig. 6A, purple dots, and Supplementary Data 12), making a total of 43 enriched TFs. Of these, CEBPB and SPI1 were enriched at cQTLs based on both predicted and experimentally determined binding sites. By running the enrichment analysis on the whole ReMap ChIP-seq database, we also confirmed the cell type specificity of TF binding at cQTLs, with nearly all enriched TFs occurring in monocytes or other myeloid cell types (Supplementary Fig. 7A and Supplementary Data 12).

Overall, nearly a third of cQTLs (205/641, 31%) were predicted to be bound in monocytes by at least one out of the 716 TFs included in the analysis. We next examined the potential allelic impact of cQTL variants on these TF binding events. For this, we took advantage of two deep learning frameworks, DeepSea[51] and Enformer[51,52]. These models were pre-trained on public genomic datasets, enabling us to infer the effects of sequence variation on signals in these training data. We focused on predicting ChIP-seq signals for sequence-specific TFs bound at cQTLs, with data for 77/205 and 179/205 cQTL-bound TFs available in the DeepSea and Enformer training sets, respectively (Supplementary Data 13 and Supplementary Fig. 7B). To ascertain perturbation effects, we compared the magnitude of DeepSea- and Enformer-predicted perturbations at cQTLs with those detected for randomly sampled variants across the genome, considering the top 1% perturbation score rank for each TF and tool as the signal threshold (see 'Methods'). Overall, 87 cQTLs were predicted by at least one tool to perturb the binding of at least one TF (Fig. 6B). The consistency of DeepSea and Enformer predictions, however, varied depending on the TF (Supplementary Fig. 7C). Therefore, to increase the robustness of this analysis, we focused on consensus predictions from both tools,

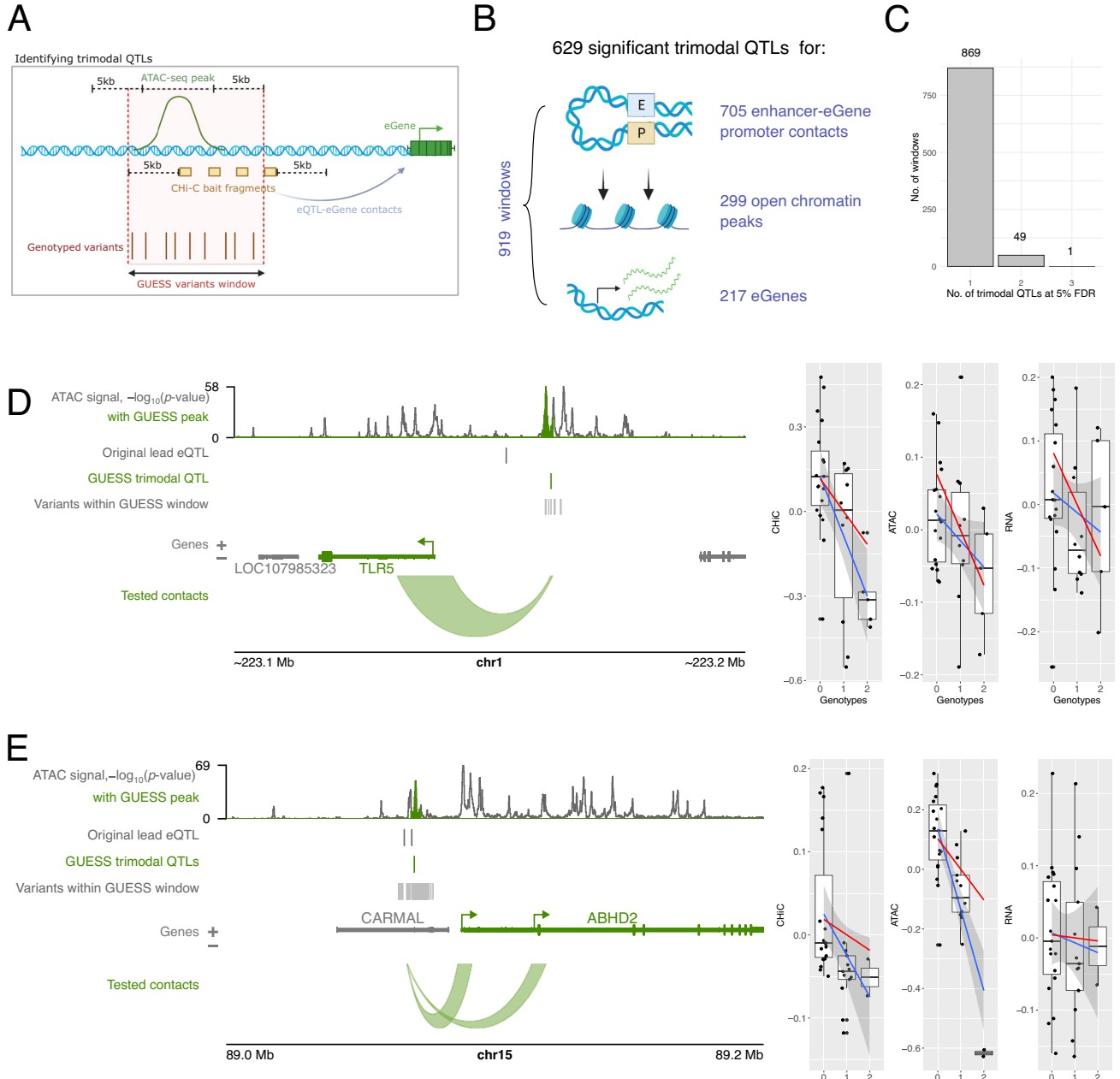

**Fig. 4 | Detection and examples of trimodal QTLs. A** Strategy for detecting tri-modal QTLs using GUESS. Regions within 5 kb of ATAC-seq peaks and CHi-C *DpnII* bait fragments were identified, and all genotyped variants were queried within these regions. Created in BioRender: https://BioRender.com/f51r706. **B** Overview of the significant findings from the GUESS analysis. Created in BioRender: https://BioRender.com/w52b348. **C** Number of trimodal QTLs at 5% FDR that best explained the observed phenotypes in each window. **D**, **E** Examples of GUESS loci where the best model (combination of genetic variants with the largest marginal likelihood score) contained a single trimodal QTL (also significant at 5% FDR) that was associated with chromosomal contact with the eGene (*TLR5* and *ABHD2*,

respectively), chromatin accessibility (highlighted in green in ATAC-seq track) and eGene expression. ATAC-signal is shown as $-\log_{10}(p$ value) pileups determined by Genrich[140]. Boxplots show the genetic effects of the variant on each modality (boxes show 25th, 50th and 75th percentiles, with upper and lower whiskers to the largest or smallest value no further than 1.5 x the interquartile range from the hinge). The red lines represent the regression lines based on the posterior mean of the regression coefficients of the GUESS model, and the blue lines represent the Maximum Likelihood Estimation (MLE) with a 95% confidence interval. Panels (**D**) and (**E**) were plotted using the Plotgardener R package[125]. Source data for **C**–**E** are available on OSF[168].

resulting in 181 perturbed binding events of 50 TFs by 49 cQTLs (Fig. 6B). Just under half of these cQTLs (24/49) were predicted to perturb the binding of multiple TFs (Fig. 6B), which is consistent with TF cooperativity at enhancers[53–55]. TFs whose binding was perturbed by cQTLs included the myeloid regulators SPI1/PU.1, CEBPB, STAT3/5A, IKZF1 (Ikaros), and NFIC, as well as JUND/FOS that jointly form the AP-1 complex (Fig. 6C and Supplementary Data 13). Notably, only two cQTLs were predicted to perturb CTCF binding: the rs7146599 contact

eQTL and the rs2353678 trimodal QTL, both of which are known CTCF binding QTLs detected in multi-individual ChIP analyses[56,57].

The binding of three TFs (SPI1, CEBPB and STAT3) was predicted to be perturbed by more than ten cQTLs each. The cQTLs predicted to perturb SPI1 binding were highly enriched for known SPI1 tfQTLs[57] compared with the rest of cQTLs (5/19 vs. 29/622 at local FDR < 0.05, accounting for LD [$r^2 > 0.99$]; Fisher test $p$ value = 0.001). Overall, just under two-thirds (31/49, 63.3%) of cQTLs predicted to perturb the

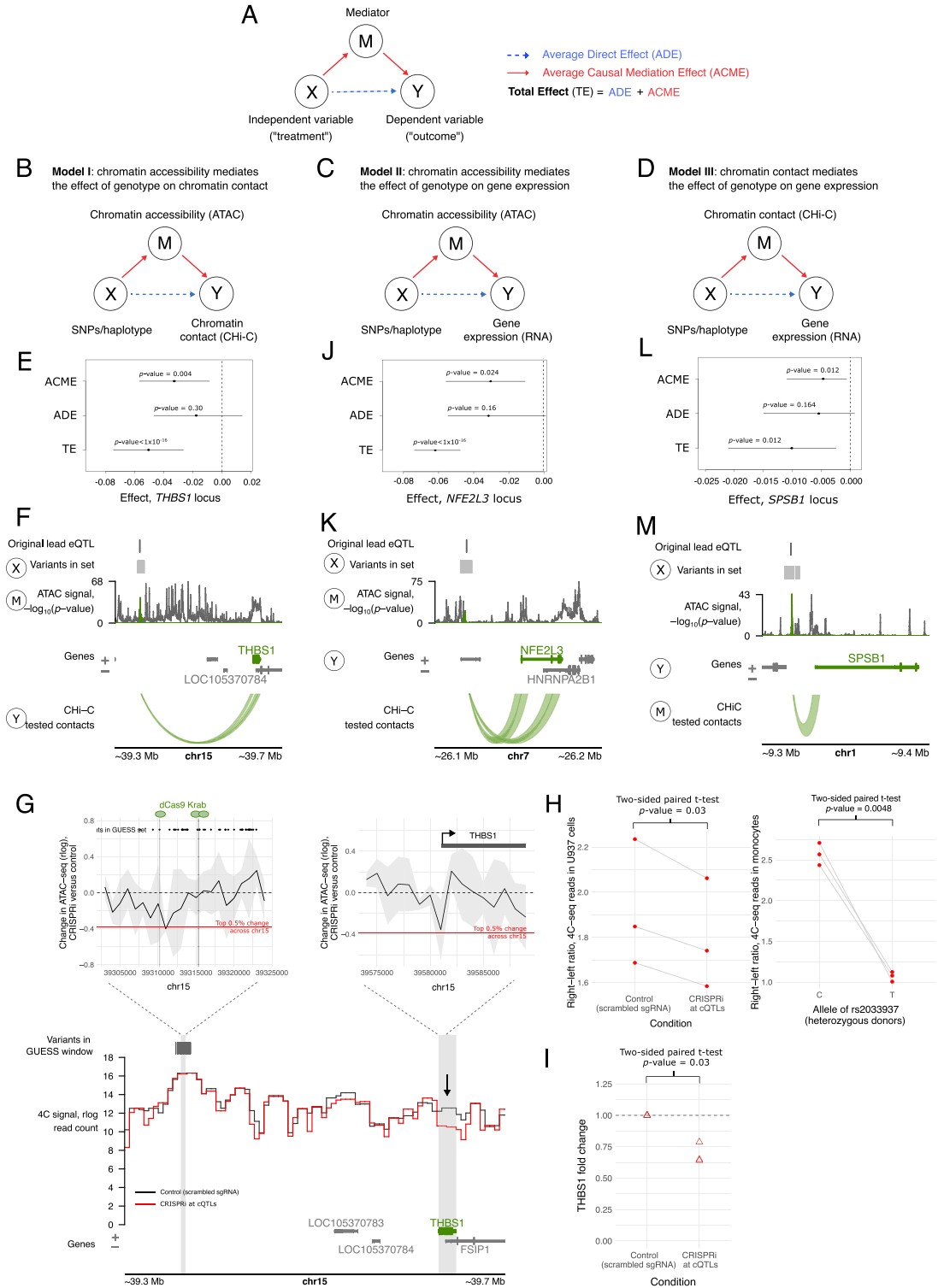

binding of at least one TF affected the predicted binding of SPI1, CEBPB and/or STAT3, either alone or in combination with other TFs. However, only a minority of cQTLs directly disrupted the recognition motifs of these factors, while others disrupted either the recognition motifs of other TFs whose binding they were also predicted to perturb or no known motifs (Fig. 6D, top and Supplementary Data 13). For example, only one out of the 12 cQTLs predicted to perturb STAT3 binding disrupted the canonical STAT3 motif, while another 5/12 cQTLs disrupted the recognition motifs of other bound TFs, including CEBPB/D,

JUNB and KLF9 (Fig. 6D, bottom). Jointly, these results suggest that the effects of cQTLs on TF binding can be cooperative or indirect, potentially mediated by perturbations in the binding sites of either known or not yet identified binding partners of these TFs.

## cQTLs associate with healthy and pathological genetic traits

As a first indication of the effects of cQTLs on physiological traits and disease, we examined the representation of these variants in the GWAS catalog[58]. Overall, ~36.5% of cQTLs (234 out of 641, including 3 out of 19

**Fig. 5 | Evidence for causal relationships at trimodal QTLs. A** The causal mediation strategy. **B–D** Overview of the three types of models considered. **E, F** Example of Model I with full mediation. Plot **E** summarises the Average Causal Mediation Effect (ACME), Average Direct Effect (ADE) and Total Effect (TE) (dot−mean effect, lines−95% bootstrap confidence intervals; non-significant if spanning 0; two-sided $p$ values were computed using the non-parametric bootstrap procedure in the R package mediation[158]). The ADE confidence interval spans zero, indicating full mediation. Plot (**F**) shows the three modalities in the mediation model: SNPs/haplotype, i.e. the genetic variants in the GUESS set $X$, the ATAC-seq signal as a mediator $M$ and chromatin contact with the eGene promoter as the outcome $Y$. **G** CRISPRi at the cQTLs in the *THBS1* locus. Top left: Change in ATAC-seq signal at the location of CRISPRi perturbation (grey lines show dCas9-KRAB target regions). The black line shows the change in ATAC-seq in CRISPRi U-937 cells with locus-targeting versus non-targeting gRNAs (rlog reads, mean of three

biological replicates), grey ribbon represents standard deviation and red line shows top 5% change across a 2 Mb window. Top right: CRISPRi-induced change in ATAC-seq at the canonical *THBS1* promoter. Bottom: CRISPRi-induced change in mean 4C-seq signal ($N = 3$ per condition). Vertical grey bars show the viewpoints at the cQTLs and *THBS1*; the black arrow highlights 4C-seq signal at *THBS1* promoter (difference not statistically significant, FDR-adjusted $p$ value = 0.66). **H** Left plot: a global shift in contact directionality from the cQTL region within a 2 Mb window. Right plot: the shift observed in allele-specific 4C seq in primary monocytes (three heterozygotes for cQTL rs2033937). **I** qPCR-detected fold change in *THBS1* expression in CRISPRi cells versus control cells ($N = 3$). The $p$ value is from a two-sided, paired *T*-test on ΔCt values (Supplementary Fig. 6C). **J–M** Examples of full mediation in Models II and III, respectively, similar to (**E**, **F**). **F**, **G**, **K**, **M** used the Plotgardener R package[125]. Source data for **E**, **G–J**, **L** are available on OSF[168].

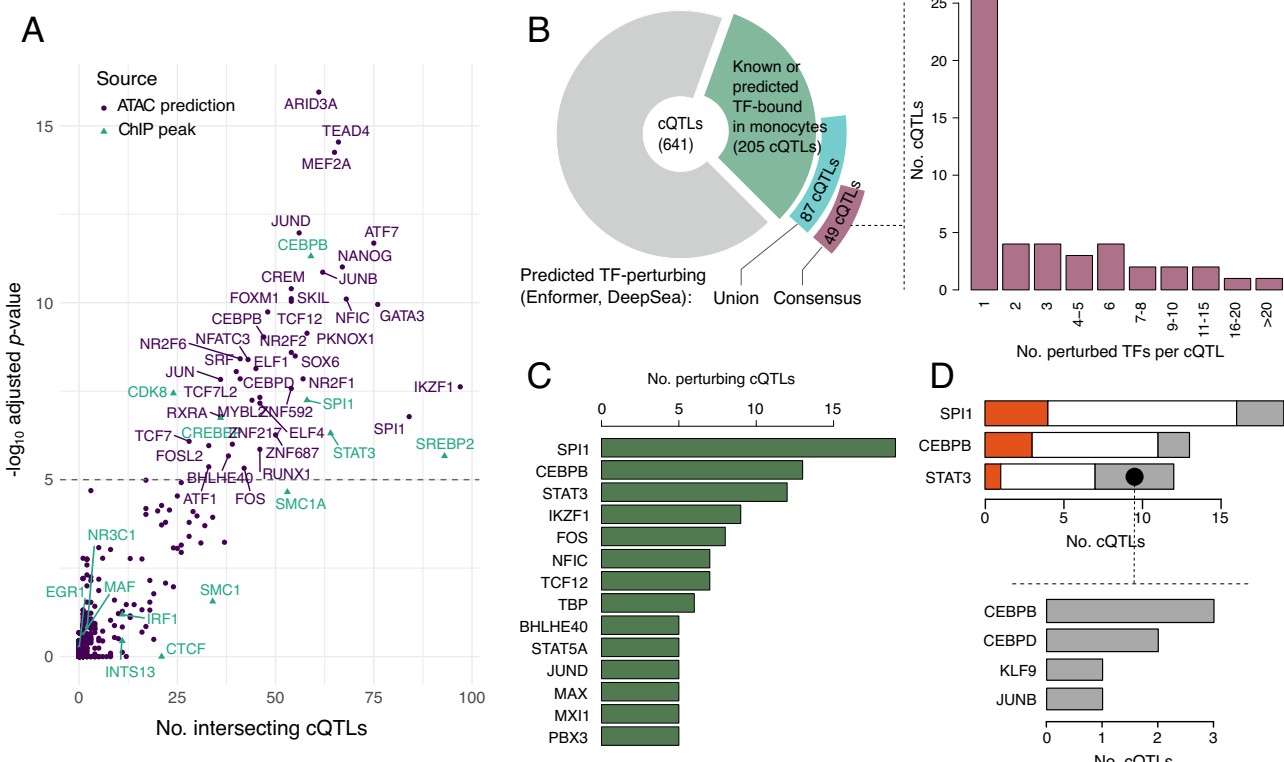

**Fig. 6 | Effects of contact QTLs on TF binding. A** Significantly enriched TFs at cQTLs using monocyte ChIP-seq peaks (ReMap catalogue and in-house CTCF, green triangles) and ATAC-seq predicted binding (union of peaks from MaxATAC and footprints from TOBIAS, per TF, purple dots). TFs with adjusted $q$ value > 5 from Remapenrich are labelled, indicating a $p$ value < 0.05 after adjusting for multiple testing. All ChIP-seq TFs are labelled for reference. The source data for significant TFs can be found in Supplementary Data 12. **B** Pie chart: cQTLs with predicted perturbations in TF binding, detected by Enformer or DeepSea. The green segment shows the number of cQTLs binding TFs. The blue and purple segments show the number of cQTLs that are further predicted to perturb the binding of those TFs,

either through the union (blue) or consensus (purple) of Enformer and DeepSea. Bar chart: histogram of the number of TFs predicted to be perturbed by cQTLs with at least one TF perturbation. **C** TFs whose binding was predicted to be perturbed by at least 10 cQTLs. **D** Top: numbers of cQTLs with predicted effects on the binding of SPI1, CEBPB and STAT3 that disrupted the known sequence binding motif for either the same (red) or other predicted perturbed TFs (grey). Bottom: TF motifs disrupted by cQTLs that were predicted by Enformer to perturb STAT3 binding jointly with other TFs but did not disrupt the known STAT3 motif. Source data for **A–D** are available on OSF[168].

BaseQTL-detected contact eQTLs) overlapped the reported GWAS variants for a total of 304 traits accounting for LD ($r^2 > 0.8$; Supplementary Data 14), with 56 cQTLs directly coinciding with these reported GWAS variants (Fig. 7A). Annotation of the associated GWAS traits based on EFO ontology[59] revealed terms such as 'haematological measurement' and 'leucocyte count' among the top trait categories (Fig. 7B), implicating monocytes as a likely causal cell type.

As an example, 29 cQTLs overlapped with genetic loci for the trait 'white blood cell count', implicating target eGenes such as nuclear receptor corepressor 1 (*NCOR1*, located ~200 kb away from the

trimodal QTL rs9910148) proposed as a key immunometabolic regulator[60], the lysosomal gene *LAMP1* associated with mononuclear phagocyte activation[61] (~34 kb away from the trimodal QTL rs9604045), as well as EP300-interacting inhibitor of differentiation 2 (*EID2*, ~96 kb away from the trimodal QTL rs1865092). We next sought to confirm the overlap of the causal cQTL and GWAS signals using formal colocalisation analysis. Since GUESS summary statistics are unsuitable for this approach, we leveraged the fact that all cQTLs are known eQTLs by design. Querying the Open Targets database[62], we confirmed the colocalisation of the published eQTL signals for these

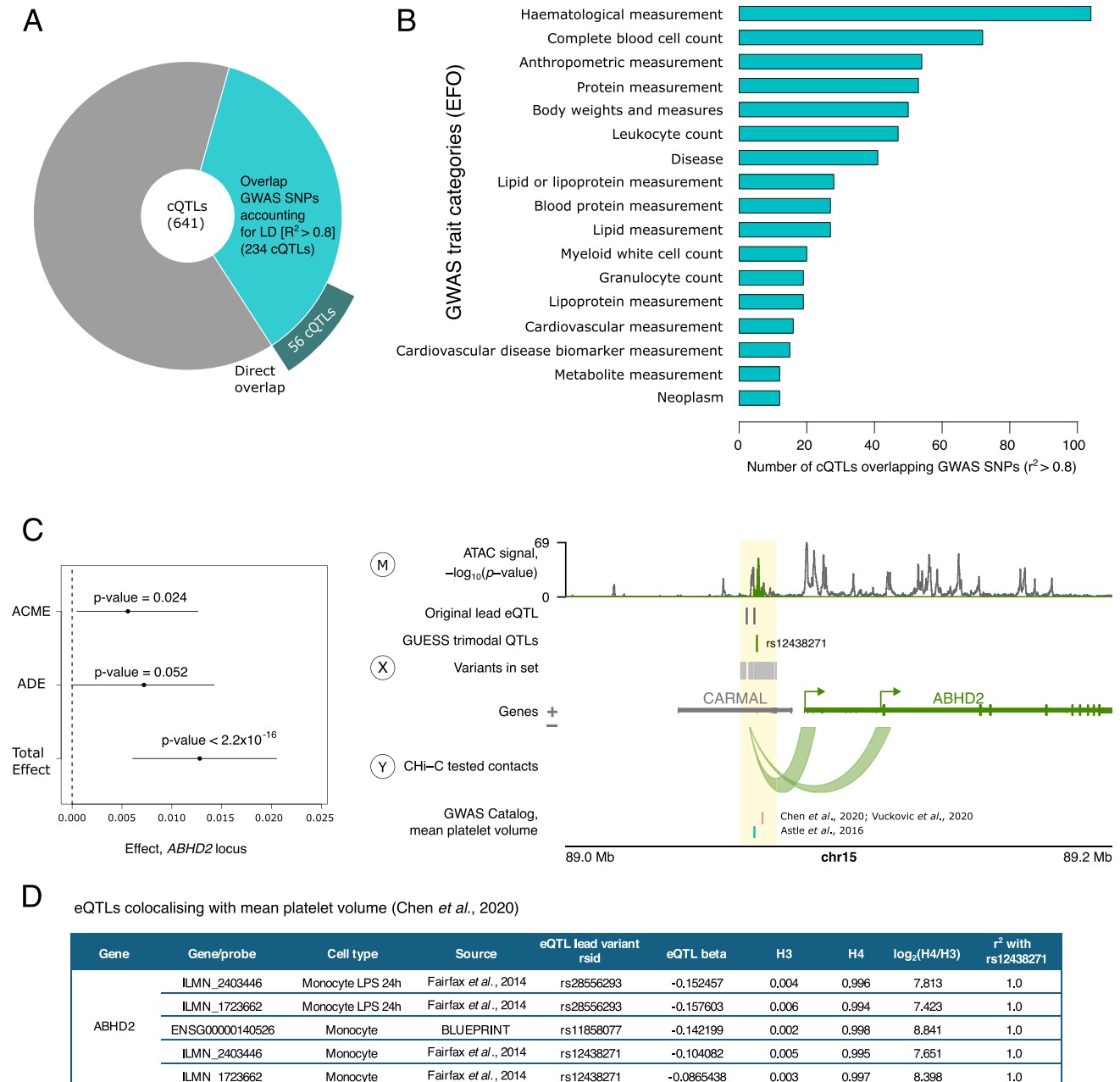

**Fig. 7 | Trimodal QTLs overlap variants associated with healthy and pathological traits. A** Pie chart showing the number of cQTLs intersecting GWAS loci through LD (light blue) or the same variant (dark blue). **B** Number of cQTLs in each of the GWAS trait categories from the Experimental Factor Ontology (EFO). **C** Example of a trimodal locus with evidence for causal mediation associated with a human trait: mean platelet volume. The forest plot on the left shows the result of the mediation analysis, summarising the three effects (ACME, ADE and Total Effect; (dot—mean effect, lines—95% bootstrap confidence intervals; non-significant if spanning 0; two-sided $p$ values were computed using the non-parametric bootstrap procedure in the R package mediation[158]). Since the ADE confidence interval spans 0 in this case, this is an example of full mediation. On the right, the locus plot at *ABHD2* shows the intersection between the trimodal QTL locus and the GWAS locus for mean platelet volume (yellow highlighted region). The letters in circles represent the three modalities in the mediation model: SNPs/haplotype, i.e. the genetic variants in the GUESS set (treatment, *X*), the ATAC-seq signal (mediator, *M*) and chromatin contact with the eGene promoter (outcome, *Y*). ATAC-signal is shown as $-\log_{10}(p$ value) pileups determined by Genrich[140]. **D** Evidence from Open Targets Genetics[62] for eQTL signals for *ABHD2* in monocytes colocalising with the GWAS signal for mean platelet volume[67]. The H3 value shows the posterior probability of two different causal variants, and H4 is the posterior probability of one causal variant, with the $\log_2$ ratio showing the posterior probability evidence for versus against shared causal variants. The final column shows the LD, as a measure of $r^2$, between the lead eQTL variant and the trimodal QTL, rs12438271. The locus plot in (**C**) was generated using the Plotgardener R package[125]. Source data for **A**–**C** are available on OSF[168].

variants with the white blood cell traits in monocytes ($N = 10$) or blood cells more broadly ($N = 21$) at all 22 cQTLs overlapping GWAS SNPs with available GWAS summary statistics required for this analysis (see 'Methods' and Supplementary Data 14).

In addition to baseline blood cell phenotypes, we also observed overlap with genetic signals for inflammatory traits. For example, the *LAMP1*-associated rs9604045 and another trimodal QTL (rs7488791, associated with the expression of a scavenger receptor *SCARB1*,

located ~33 kb away) were also GWAS variants for the levels of C-reactive protein, a well-established biomarker of inflammation[63]. Two trimodal QTLs (rs4389574 and rs4698412) were associated with Parkinson's disease and implicated bone marrow stromal cell antigen 1 (*BST1*, 21–28 kb away), which has a known role in humoral inflammatory response[64]. This is notable given the emerging role of the innate immune system in this disorder[65].

Finally, we focused on the abhydrolase domain containing 2, acylglycerol lipase (*ABHD2*) locus, where the trimodal QTL rs12438271 overlapped signals for mean platelet volume, falling within the 95% credible set in three studies[66–68] (Fig. 7C). Interestingly, in this locus, chromatin accessibility fully mediated the effect of genotype on chromosomal contact (mediation Model I; TE *p* value < 10⁻¹⁶, ADE *p* value = 0.052, ACME *p* value = 0.024) (Fig. 7C; sensitivity shown in Supplementary Fig. 6F). There was strong evidence of colocalisation between the GWAS signal and eQTL signal in monocytes, according to Open Targets[62] (Fig. 7D), suggesting a mechanism of action of these trait-associated variants on platelet function, over and above the information provided by eQTL studies.

Jointly, these examples highlight the relevance of enhancer-promoter contacts in mediating the effects of genetic variants associated with healthy and pathological traits.

### A contact eQTL affects *PCK2* expression by modulating CTCF binding and chromatin insulation

Contrary to the majority of BaseQTL-detected contact eQTLs that showed a consistent direction of allelic effect on gene expression and promoter contact, variants identified in the *PCK2* and *C7orf50* loci showed the opposite direction of effects on these two properties (Fig. 3C). Notably, neither of these variants associated with active histone marks, and the *PCK2* contact eQTL variant, rs7146599, also did not associate with chromatin accessibility. This QTL is a known CTCF tfQTL[56], was predicted by both Enformer and DeepSea to perturb CTCF binding in vivo and affects the canonical CTCF binding motif (Fig. 8A). To further validate the allelic effect of rs6477612 on CTCF binding, we performed ChIP-seq for CTCF in monocytes from three individuals who were heterozygous for rs7146599. This confirmed the presence of a CTCF peak at rs7146599, with the reference allele (G) binding 12-fold more CTCF than the alternative (A) allele (Fig. 8B). Furthermore, we confirmed by allele-specific 4C-seq in the same individuals that the alternative allele of this cQTL was associated with a weaker contact with the *PCK2* gene than the reference allele (Fig. 8C).

In contrast to the decreased contact with *PCK2* and predicted decreased CTCF binding, the alternative allele of rs7146599 is associated with increased *PCK2* expression across multiple eQTL studies in monocytes[32,44,69] and whole blood[3,31,44,70,71] (Fig. 8D). While we did not find a significant association between rs7146599 and *PCK2* expression in our cohort, we observed the same direction of effect (allelic fold change = 1.05). We therefore asked whether the *PCK2* promoter contacted additional enhancers in the alternative genotype. To address this, we compared the patterns of *PCK2* promoter interacting regions between donors that were homozygous for the reference or alternative allele of rs7146599. In the homozygous alternative genotype, we observed increased interactions between the *PCK2* promoter and regions upstream of rs7146599 (CHiCAGO score ≥ 5, 5 kb resolution) containing open chromatin and predicted enhancer elements in monocytes (based on data from Ensembl regulatory build[72]) (Fig. 8E). Notably, both rs7146599 and the *PCK2* promoter share the same TAD and do not intersect a TAD boundary, based on our monocyte Hi-C data (Supplementary Data 5); thus the observed effects likely take place at a sub-TAD level.

Jointly, these data suggest that rs7146599 perturbs the function of a CTCF-dependent insulator element that shields *PCK2* from upstream enhancers (Fig. 8F).

## Discussion

In this study, we used multimodal profiling of molecular phenotypes across individuals to establish the effects of genetic variants at enhancers on enhancer activity, connectivity and gene expression in human primary monocytes. We observed widespread shared genetic effects of known eQTL variants across these three modalities, indicating a common underlying molecular mechanism that appears to be independent of the specific transcription factors whose binding to enhancers is disrupted by these variants. In addition, we also found an example whereby a distal variant likely influences gene expression through perturbing a CTCF-dependent chromatin insulator and the resulting co-option of downstream enhancers.

Our choice of eQTLs as anchors for studying the relationship between enhancer activity and connectivity was motivated by the fact that enhancers can efficiently buffer the effects of genetic variation, even when this variation results in detectable effects on transcription factor binding and chromatin state[73,74]. Furthermore, genes are commonly controlled by multiple enhancers with a considerable degree of redundancy[75,76]. Consequently, disruptions in enhancer activity may not lead to prominent alterations in target gene expression, especially in stable conditions[77,78]. Therefore, we focused on variants that have demonstrated effects on gene expression in unstimulated primary monocytes to prioritise cases where the observed associations between these variants and enhancer activity and connectivity are more likely to be functionally meaningful. Consistent with this expectation, many identified QTLs have known associations with diverse genetic traits and diseases, including those directly implicating monocytes.

Assessing the effects of genetic variants on chromosomal contacts by multi-individual chromosome conformation capture analyses presents inherent challenges, particularly in a high-throughput setting and for longer-range contacts. The large numbers of sequencing reads per donor required to resolve individual contacts and the complexity of the experimental protocols hamper the statistical power of this analysis in terms of both feasible sequencing coverage and cohort size, limiting the sensitivity of Hi-C-based association studies[79–81]. We and others have previously used Promoter Capture Hi-C in small-scale analyses of allelic effects on enhancer-promoter contacts[40,57]. Here, we targeted the Capture Hi-C system to short restriction fragments containing eQTL variants in addition to their target eGene promoters, which ensured that the sequencing reads directly encompassed the variants of interest. This enabled us to conduct within-individual allelic analysis for heterozygous donors in addition to genotype-based QTL detection across all individuals. Our approach is complemented by two parallel independent studies that have employed Hi-C and HiChIP across individuals, respectively, to determine the effects of genetic variation on chromosomal contacts in T lymphocytes[81,82].

Our study is based on a relatively small cohort of 34 donors, which is only slightly larger than that used in the HiChiP-based study[82]. It was challenging to expand this cohort due to the limited scalability and high labour intensity of the Capture Hi-C assay. We therefore focused on devising statistical methodologies that can detect robust genetic associations in Capture Hi-C data with increased power. Adapting our recently developed Bayesian QTL detection method BaseQTL to Capture Hi-C analysis, we were initially able to detect 19 variants associated with eQTL-promoter contact, as well as a multitude of ATAC QTLs. The observation that most variants associated with promoter contact were also ATAC QTLs, as well as eQTLs by design, motivated us to pursue the detection of shared genetic effects across all three of these modalities at increased power. Our multi-modality Bayesian approach identified 629 putative trimodal QTLs, mirroring the well-known advantages of multivariate analysis of variance (MANOVA) for detecting patterns between multiple correlated dependent outcomes over the univariate analysis (ANOVA) that tests one outcome at a time[83]. We observed a

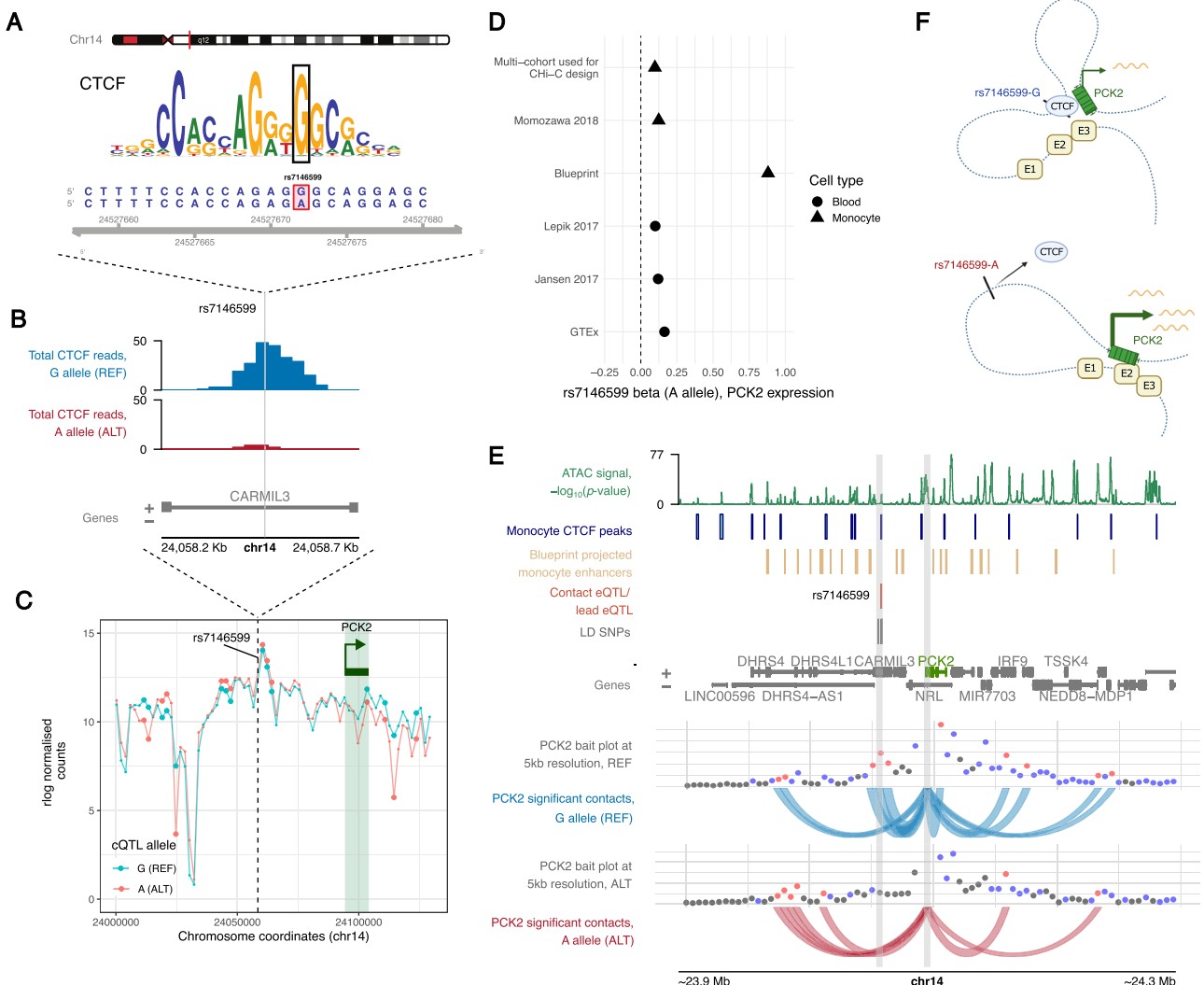

**Fig. 8 | A contact eQTL perturbs CTCF binding and chromatin insulation.**
**A** Position of rs7146599 within the CTCF motif. **B** ChIP-seq reads for CTCF intersecting the reference allele (G, blue) or alternative allele (A, red) of rs7146599 in heterozygous reads. The pileup shows the total of paired-end reads (read 1 and read 2) intersecting the cQTL, pooled across three heterozygous individuals. **C** 4C-seq validation of rs7146599-*PCK2* allelic looping in three heterozygous individuals. The 4C-seq viewpoint is shown by the dashed line at rs7146599, and the rlog normalised 4C-seq reads (mean across three individuals) are shown for the reference allele (G, blue) or the alternative allele (A, red). Larger dots indicate significantly different interactions between alleles (4Cker statistical test *p* value < 0.05 after adjusting for multiple comparisons). **D** eQTL effects of rs7146599 on *PCK2* across multiple cohorts. Betas with respect to the alternative allele (A) are shown in studies of monocytes (the monocyte multi-cohort analysis used to design the eQTL CHi-C

experiment[32], Momozawa et al.[69] and Blueprint[44]) and whole blood (Lepik et al.[71], Jansen et al.[70] and GTEx[3]). **E** Effect of genotype on contact profiles in the locus. Individuals were split into monozygous reference (G allele, *N* = 11) or alternative (A allele, *N* = 9) genotypes and merged in CHiCAGO to produce an average number of counts in bins of 5 kb. Contact profiles are shown from the viewpoint of the *PCK2* promoter, with the location of the promoter and the contact QTL highlighted with grey rectangles. The monocyte CTCF peaks shown in this figure were generated using the ChIP-seq data shown in (**B**). ATAC-signal is shown as −log$_{10}$(*p* value) pileups determined by Genrich[140]. **F** Proposed mechanism schematic showing how perturbed CTCF binding at rs7146599 could affect the insulation of the *PCK2* promoters from distal enhancers, denoted as E1, E2 and E3. Created in BioRender: https://BioRender.com/c07a901. Source data for **B**–**E** are available on OSF[168].

strong enrichment for concordant directions of effect across all three modalities but also detected abundant shared associations showing discordant effect directions, consistent with the findings of the recent Hi-C-based association analysis[81]. While the effect sizes at such discordant associations were generally lower than at those showing full concordance, this observation points to the complexity of regulatory relationships between the chromatin state, chromosomal looping and gene expression.

We followed up our observations of shared effects by causal mediation analysis and CRISPR interference to demonstrate that enhancer activity and/or accessibility can causally mediate enhancer connectivity, and both can mediate gene expression at a subset of trimodal QTLs. It is likely, however, that the true number of variants

exhibiting shared effects on these three modalities, as well as causal relationships between them, is even larger. While increasing the cohort size of multi-individual chromosome conformation analyses remains a clear aspiration for future research, we hope that the statistical approaches presented here will be useful for investigating the genetic effects on other molecular phenotypes, particularly those that are challenging to profile in large cohorts.

Our computational predictions using an ensemble of sequence-based deep learning tools suggest that many trimodal QTLs affect the binding of one or more TFs, including the myeloid transcription factor SPI1/PU.1 implicated in enhancer-promoter looping[28,57,84]. These results point towards an inherent genetic coupling of enhancer activity and connectivity that is mediated by CTCF-independent mechanisms and

may be largely independent of the identity of specific TFs recruited to enhancers. Our results are consistent with the findings of a recent independent analysis showing that variants affecting coordination between cis-regulatory modules affect the binding of a diverse range of lineage-specific TFs[85].

Which molecular mechanisms could underpin the causal link between enhancer activity and connectivity? Notably, in our analysis, only two cQTLs (including one special case discussed below) perturbed the binding of the architectural protein CTCF. However, the cohesin complex, and particularly its isoform containing the STAG2 subunit, is known to be recruited to active enhancers, including those that do not bind CTCF[86,87]. This recruitment could be directly impacted by genetic aberrations in enhancer activity. Mechanisms of cohesin recruitment to active chromatin are not fully understood but may involve interaction with the H3K4me1 histone mark[88] or with acetylated histones[89]. However, a significant number of enhancer-promoter contacts, particularly those in the shorter distance range, are likely independent of both cohesin and CTCF[23–26]. It is possible that, in these cases, enhancer-promoter proximity is facilitated by phenomena such as liquid-liquid phase separation[15,90–92] that can be directly mediated by the recruited TFs[93–95] and core cofactors, including the Mediator complex[94,96–99]. Finally, transcription itself is known to play a role in chromatin looping[100,101], and while other mechanisms are likely necessary for initiating enhancer-promoter contacts upon gene induction, recent evidence suggests a role for RNAP II and transcription in this process[102,103].

Finally, we show that in the *PCK2* locus, direct disruption of the canonical CTCF sequence recognition motif by a genetic variant leads to perturbed promoter contact and increased distal gene expression without an apparent effect on chromatin activity *in cis*. The influence of genetic perturbations of CTCF binding on 3D chromosomal architecture is expected and has been reported previously[80,104]. It is also known that disruption of enhancer-promoter contacts, including through perturbations of architectural proteins, can drive down gene expression[23–25]. However, in the specific case we have identified, genetic perturbation of a CTCF binding site and ablation of its contact with *PCK2* promoter instead led to increased expression of *PCK2*. We propose that this effect can be attributed to the potential co-option of enhancers located further upstream of the *PCK2* promoter, facilitated by the removal of a CTCF-mediated chromatin boundary. This model is consistent with the classic function of CTCF at insulator elements[105,106] and with instances of enhancer hijacking resulting from dissolved chromatin boundaries in ectopic settings[23,107–109] and in cancer[2,89,110]. In addition, CTCF has been recently shown to mediate the minority of tissue-invariant promoter-enhancer contacts that persist independently of enhancer activity[111]. Therefore, genetically determined CTCF binding, either *in cis* or in trans to enhancers, may be a key mechanism for uncoupling enhancer activity and promoter communication. However, the apparent rarity of such direct effects on architectural protein binding reinforces the notion that, at least in the case of communication between active enhancers and promoters, genetic effects on chromosomal contact are more commonly mediated by epigenetic phenomena.

In conclusion, our analysis reveals abundant shared effects of common genetic variants on enhancer activity, connectivity and gene expression in human primary cells and highlights the ability of genetic variants to influence gene expression through direct modulation of architectural protein binding. Taken together, our results provide insights into the mechanisms of enhancer-driven gene control and their genetic perturbation in disease.

## Methods
Research performed in this study complies with all relevant ethical regulations. Human monocytes were extracted from apheresis cones

discarded after platelet donation by NIHR National BioResource volunteers after obtaining informed consent and ethical approval (12/EE/0040, East of England-Hertfordshire Research Ethics committee). Since sex is known to affect monocyte biology[112–115], we sought to recruit a single-sex cohort. The male sex was chosen based on platelet donor availability at the recruitment centre. Sample-level information is provided in Supplementary Data 15. Donor sex was confirmed by genotyping.

### Monocyte eQTL data
To inform our CHi-C design, we used a multi-cohort eQTL analysis of unstimulated monocyte data from a total of 1480 samples[32]. The samples for this analysis comprised individuals from three cohorts: (1) The Wellcome Trust Centre for Human Genetics (WTCHG), University of Oxford, UK ($N = 432$); (2) The CEDAR project, University de Liege, Belgium ($N = 300$) and (3) The Cardiogenics Consortium eQTL project, University of Leicester and University of Cambridge ($N = 758$)[31]. The data from this study was subsequently incorporated into the eQTLGen Consortium as a validation of monocyte *trans*-eQTLs[31]. Briefly, significant eQTLs (FDR < 0.01) were detected using a linear mixed model (LIMIX[116] version 0.8.5, PEER version 1.0[117]) and only variants within the gene body or 1 Mb windows flanking the 5' and 3' of the tested gene were considered. Independent signals were identified using forward selection, which recursively adds the strongest associated variant of the previous association test as a covariate. Up to five iterations of forward selection were performed, thereby identifying up to five independently associated variants with the expression of each eGene[32].

### Human *DpnII* eQTL capture Hi-C design
We designed an eQTL capture Hi-C system, enabling us to assay chromatin loops containing enhancer-borne eQTLs at the loop anchors. We found all SNPs in tight LD ('proxy SNPs') with the lead monocyte eQTLs from the analysis described above ($R^2 >= 0.9$) using PLINK[118], with 1000 Genomes[119] phase 3 (European samples) at MAF 10%. We then located the transcription start sites of the corresponding eGenes (Ensembl Genes release 96) using the Biomart R package and removed all proxy SNPs that were within 10 kb of a TSS for the corresponding eGene. SNPs and TSS were assigned to *DpnII* fragments, and we only retained eQTLs and proxy SNPs that were within 100 bp of a *DpnII* cut site to increase our chances of detecting these SNPs within our CHi-C reads. To enrich our design for SNPs with likely regulatory function, we removed proxy SNPs that did not intersect a known regulatory region (Ensembl Regulatory Build[72] release 96) whilst keeping all remaining lead eQTLs in the design. We also removed 31 regions that had very large LD blocks (>200 kb). Next, focusing on gene promoters, we generated 50 kb 'eGene windows' that were situated from 5 kb downstream of the eGene promoter to 45 kb upstream, with respect to the position of the eQTL LD block. We targeted all gene TSS within these 50 kb windows. Likewise, we generated a 50 kb 'mirror window' to the opposite side of the LD block and targeted all TSS within that region (Fig. 1A). Thus, we targeted a set of distance-matched control genes for each eQTL-eGene pairing. We designed 120 bp probes to capture the *DpnII* fragments encompassing the SNPs and proxies (one probe per fragment). In total, our capture system contained 18,178 probes that targeted 1458 eQTLs and 2571 proxy SNPs (a mean of 3 SNPs per eQTL) and 8533 genes. Furthermore, since we had captured eGene promoters in our study, we were able to include eQTLs and/or proxy SNPs at 'other ends' of CHi-C contacts in our analysis. We also directly captured 4718 distance-matched control genes for these eQTL loci. Note that CHi-C data arising from this array design was further filtered post-hoc prior to detection of contact QTLs, as described below in 'CHi-C data processing for contact QTL detection'.

## Monocyte purification, crosslinking and lysis

Peripheral blood mononuclear cells were isolated by standard Ficoll gradient centrifugation, and classical monocytes (CD14+, CD16−) were isolated using EasySep™ Human Monocyte Isolation Kit from Stemcell Technologies according to the manufacturer's instructions. For each sample, 2 million monocytes were frozen in TRIzol and stored at −80 °C for later RNA extraction. Meanwhile, between 16.4 and 64.8 million (median 25.3 million) monocytes were crosslinked in 2% formaldehyde for 10 min, after which the reaction was quenched with 0.125 M glycine, the supernatant removed and the crosslinked cell pellet flash frozen and stored at −80 °C. Prior to processing for CHi-C, ATAC-seq and DNA extraction (for genotyping), the crosslinked pellets were thawed on ice and lysed for 30 min in Hi-C lysis buffer (10 mM Tris-HCl, pH 8.0, 10 mM NaCl, 0.2% IGEPAL CA-630 and 1X protease inhibitors) at a density of approximately 3330 cells/µL. The nuclei were then divided into aliquots of 50 K−3 million cells, centrifuged, the supernatant removed, and the pellets snap-frozen on dry ice and stored at −80 °C.

## Capture Hi-C

Capture Hi-C libraries (1–2 technical replicates per donor) were generated on 300,000 to 1 million crosslinked nuclei as previously described[120,121]. Following lysis, the nuclei were permeabilised and digested with *DpnII* (NEB) overnight. The restriction overhangs were filled in using a biotinylated dATP (Jena Bioscience), and ligation was performed for 4 h at 16 °C (T4 DNA ligase; Life Technologies). The crosslinks were reversed using proteinase K and overnight incubation at 65 °C, followed by purification with SPRI beads (AMPure XP; Beckman Coulter). Short fragments up to 1000 bp were produced via tagmentation, and the biotinylated restriction junctions were then pulled down using MyOne C1 streptavidin beads (Life Technologies). PCR amplification (5 cycles) was performed on the libraries directly bound to the C-1 beads, and the libraries were purified using SPRI beads as before. eQTL Capture was performed using the custom-designed Agilent SureSelect system described above, following the manufacturer's protocol. The libraries were sequenced using 150 bp paired-end sequencing on an Illumina NovaSeq (Novogene UK), NextSeq 500 or HiSeq 2500 (read statistics are provided in Supplementary Data 2).

## CHi-C data processing and detection of significant contacts

Where more than one sequencing run was performed for the same technical replicate, the CHi-C reads were merged using *cat* before processing further. Read processing, alignment and filtering were performed using a modified version of the Hi-C User Pipeline (HiCUP[122]) v0.7.4, HiCUP Combinations[34] (https://github.com/StevenWingett/HiCUP/tree/combinations), which first creates all possible combinations of ditags from the paired reads before mapping, then performs standard filtering for common Hi-C artefacts. At this point, alignment files for technical replicates (where present) were merged per sample. To determine the correlation between samples, we first filtered our bam files to on-target reads and converted them to .homer files using the *hicup2homer* function included with HiCUP and then created 10 kb interaction matrices using homer[123] v4.10.4. Then, pairwise stratum-adjusted correlation coefficients (SCC) were generated using the R package HiCRep[124] v1.10.0 (Supplementary Fig. 2A). To generate the deeply sequenced consensus dataset, CHi-CAGO input (.chinput) files were generated per replicate, using *bam2chicago.sh* from chicagoTools[33] v1.13. Then, read counts across all .chinput files were summed in R and CHiCAGO was run at *DpnII* resolution and at the level of 5 kb bins, as previously described[33,34]. We performed tests for enrichment at other ends of regions interacting with eQTLs and gene promoters at the resolution of 5 kb bins or *DpnII* fragments, using the *peakEnrichment4Features* function in CHiCAGO using the following datasets: histone modifications in monocytes from

ENCODE ChIP-seq data[38] (ENCFF493TKQ, ENCFF933WTT, ENCFF682RJQ, ENCFF570DGJ, ENCFF275IMY, ENCFF473HCC), classical monocyte segmentations downloaded from Ensembl Regulatory Build (http://ftp.ebi.ac.uk/pub/databases/blueprint/releases/current_release/homo_sapiens/secondary_analysis/Ensembl_Regulatory_Build/hg38/projected_segmentations) and CTCF from monocytes in Blueprint[44]. To obtain interactions within individuals, CHiCAGO was also run on each biological replicate separately. To generate locus plots using the CHi-C data and other modalities in this study, we used the PlotGardener[125] R package v1.2.10.

## CHi-C ABC

The CHi-C ABC method that we presented recently[39] is a modification of the original Activity-by-Contact (ABC) technique[41] for CHi-C data. Briefly, for a given promoter-enhancer pair, ABC score is defined as a product of Activity (accessibility and epigenetic marks) and Contact frequency (in the case of CHiC-ABC, based on normalised and imputed CHi-C signals) normalised to the sum of ABC scores for all enhancers within a five megabase window surrounding a gene's promoter. Custom scripts for running CHi-C ABC are available at https://github.com/pavarte/PCHIC-ABC-Prediction.

To select a threshold for ABC scores, we drew inspiration from a method by Xu et al.[126] based on optimising Pearson's correlation between gene-level ABC scores and the observed gene expression levels (expressed as CPMs). Specifically, we computed gene-level ABC scores by summing the products of enhancer activity and contact frequencies, including in this calculation only those enhancers that passed a specific 'enhancer inclusion' threshold on the standard (enhancer-level) ABC scores. Iterating over a range of enhancer inclusion thresholds, we then assessed Pearson's correlation between gene-level ABC scores and gene expression at each threshold for all genes. As expected, this correlation initially increased with increasing thresholds, as enhancers with baseline levels of activity and/or contact were removed from the calculation and then started to decrease as enhancers with appreciable impact on gene expression were increasingly filtered out. The standard (enhancer-level) ABC score threshold of 0.012 yielded a maximum correlation between gene-level ABC scores and gene expression ($r \sim 0.31$) and was used for downstream analyses. We additionally verified that the observed correlation was highly significant by permuting gene labels in the ABC dataset ($p$ value < 0.001 over 1000 permutations).

## Hi-C and TAD analysis

To generate TADs, Hi-C libraries from two donors (indicated in Supplementary Data 2) underwent 150 bp PE sequencing on a NovaSeq (Novogene UK). The reads were processed and filtered using HiCUP v 0.7.4, yielding ~117 and ~127.5 million clean, unique reads. TADs were called using *findTADsAndLoops.pl*, and the TADs for both replicates were then merged using *merge2Dbed.pl*, both from the homer toolkit[123] v4.10.4. We used the *foverlaps* function in data.table R package to intersect eQTLs (or SNPs in tight LD), along with their target eGene promoters, with TADs. We designated eQTLs and genes 'same TAD' if any part of the LD block for the eQTL and the gene promoter intersected the same TAD. Alternatively, we designated them 'different TAD' if both the eQTL LD block and the gene promoter intersected different TADs, and no part of the LD block was in the same TAD as the gene promoter. We disregarded loci where the eQTL and/or the gene promoter fell outside of called TADs.

## ATAC-seq

ATAC-seq libraries were generated following the omni-ATAC protocol, with modifications for crosslinked chromatin as previously described[127,128]. Between 25 thousand and 100 thousand crosslinked, lysed nuclei (1–2 technical replicates per individual) were re-

suspended in a tagmentation reaction mixture (10 mM tris HCl [pH 8.4–9.0], 5 mM MgCl2, 10% DMF, 0.005% digitonin and 0.05% tween 20 and up to 5 µL of homemade Tn5 enzyme) and incubated at 37 °C for 2 h with 1000 rpm mixing. Reverse crosslinking was performed using proteinase K overnight at 65 °C, and then samples were cleaned using minElute spin columns (Qiagen). Libraries were amplified using 5–7 cycles of PCR using KAPA HiFi DNA polymerase and purified with 1.1X SPRI (Beckman Coulter). Nucleosome profiles were manually inspected using a Tapestation with the D1000 Screen Tape system (Agilent). Libraries underwent 150 bp PE sequencing on a NovaSeq (Novogene UK) or HiSeq 2500 (read metrics given in Supplementary Data 3).

Adaptor contamination was removed from the reads using NGmerge[129] v0.3. Reads were aligned to the GRCh38 genome using Bowtie2[130] v2.2.9 using the settings: *–very-sensitive* and *-X 2000*. Next, mitochondrial reads, multi-mapped reads and non-paired reads were identified and removed using SAMtools[131] v1.3.1. For calling peaks, the bam files for technical replicates were merged (where applicable), and PCR duplicates were removed using Picard[132] v2.6.0. Post-filtering, we had a minimum of 24.9 million and a median of 34.7 million unique reads per individual. Correlation between biological replicates was determined using the *plotCorrelation* function from deepTools[133] v3.3.1 (Supplementary Fig. 2B). For each sample, we confirmed the presence of distinct fragment sizes corresponding to nucleosomes and confirmed the enrichment of nucleosome-free peaks at TSSs and mononucleosome peaks surrounding TSSs using the R package soGGi[134] v1.18.0 (Supplementary Fig. 1A–C). Fraction of reads in peaks (FRiP) scores were determined by first calling peaks on individual replicates using macs2[135] v2.2.9.1 and then counting the numbers of reads within peaks using featureCounts[136]. We combined all sample BAM files using SAMtools[131,137] *merge* to create a consensus dataset and ran the hidden Markov Model HMMRATAC[37] v1.2.10 to detect peaks of open chromatin. The intersection of these peaks with genomic features and the signal profile across genes was determined using ChIPseeker[138,139] (Supplementary Fig. 1E, F). In order to produce signal tracks for visualisation, ATAC-seq samples (individual replicates or merged bam files) were additionally processed by the Genrich peak caller[140] v0.5_dev, which produces bedGraph files of *p*-values and read pileups. To compare against other ATAC-seq datasets in human primary monocytes, we downloaded raw fastq files from studies by Weichselbaum et al.[35] (SRA Run selector: SRR9888114, SRR9888115, SRR9888116, SRR9888117, SRR9888118, SRR9888119) and Calderon et al.[36] (SRA Run selector: SRR7650767, SRR7650849 and SRR7650886). These files were processed from fastq to peaks as described for the monocyte ATAC-seq in the present study. A comparison of ATAC-seq signals is shown in the *THBS1* locus on chromosome 15 in Supplementary Fig. 1G. Intersecting peaks between the three studies (HMMRATAC[37] on merged replicates in each study) were counted using *multiinter* from BEDtools[141] v2.30.0 (Supplementary Fig. 1H).

## RNA-seq

RNA was extracted from TRIzol samples (2 million monocyte cells) using chloroform in MaXtract High-Density 2 mL tubes, followed by dissociation with QIAShredder columns and purification using the RNeasy kit with RNase-free DNase treatment (all Qiagen). RNA samples were quantified with a Qubit RNA assay (Thermo Fisher Scientific), and 800 ng RNA was submitted to Novogene UK for mRNA-seq library preparation (poly A enrichment) and 150 bp PE sequencing, obtaining ~20 million reads per sample (Supplementary Data 4). Read QC and mapping were performed according to the Novogene bioinformatics pipeline. Reads were mapped to the GRCh38 genome using STAR[142] v2.6.1d, allowing for 2 mismatches and assigned to Ensembl Release 94 genes. Pearson's correlation of gene FPKM values was determined between all pairwise samples (Supplementary Fig. 2C).

## Determining the relationship between gene expression and enhancer-promoter contacts

To probe the relationship between gene expression and enhancer-promoter contacts, we first calculated transcripts per million (TPM) for every gene in each individual. We determined significant CHi-C interactions between gene promoters and open chromatin by intersecting the peak matrix produced by CHiCAGO, which contained significant interactions at 5 kb binned resolution in each individual (CHiCAGO score ≥5), with the locations of captured gene promoters on one end and open chromatin on the other end (called by HMMRATAC in the consensus ATAC-seq dataset). We summed the number of active interacting regions per gene promoter in each individual and determined Spearman's rank correlation between log(no. of interacting regions) and log(gene TPM).

## Genotyping

DNA was extracted from 2 million crosslinked, lysed nuclei using the QiaAmp FFPE kit (Qiagen), starting from the Proteinase K step. Genotyping was performed on the Infinium Global Screening Array (Illumina) by The IoPPN Genomics & Biomarker Core Facility at King's College, London. Initial genotyping QC was performed using Zcall, and all samples passed the minimum call rate of 95%. Before imputation, variants were filtered to 5% minor allele frequency (MAF) and 100% call rate in our cohort. We removed SNPs deviating from Hardy-Weinberg equilibrium at a threshold of 1E−05. The remaining 447,026 variants underwent genome-wide imputation on the Michigan Imputation Server[143] v1.2.4 using the Haplotype Reference Consortium[144] with the following programmes: minimac4[145] v1.0.2 and phasing with Eagle[146] v2.4. Post-imputation, the variants were filtered for $R^2 \geq 0.3$ and 5% MAF, resulting in a total of 5,380,837 SNPs in our cohort. We retained the information on imputation probabilities and incorporated them into the BaseQTL analysis.

## Validation of meta-analysis eQTLs effects in our analysis cohort

To ascertain the concordance between the eQTL effects in the multi-cohort eQTL study and our monocyte cohort, we detected eQTLs in our RNA-seq data using two different approaches: (1) Matrix eQTL[147] and (2) BaseQTL[43]. For Matrix eQTL, total gene counts were determined using FeatureCounts[136] v1.5.0-p3 with default settings in the Novogene analysis pipeline. We used the R package DESeq2[148] v1.26.0 to merge the counts from RNA-seq technical replicates (S026T6 and S02699) and normalise across samples using *estimateSizeFactors*. Matrix eQTL[147] was run using sample genotypes. For BaseQTL, raw RNA-seq reads were processed using the full BaseQTL input Snakemake pipeline for RNA-seq data (available from https://gitlab.com/evigorito/baseqtl_pipeline/-/tree/master/input), which performs mapping using STAR[142], generates allele-specific counts using phASER[149] and determines the baseline allelic imbalance (AI) of each variant, which is then incorporated in the BaseQTL analysis. BaseQTL exploits allele-specific expression (ASE) to increase power. However, homozygous individuals erroneously genotyped as heterozygous using DNA would overestimate the allelic imbalance observed from RNA-seq reads, which may lead to false positive calls. To mitigate this source of error, we employed two strategies. First, we genotyped exonic SNPs from RNA-seq reds using BBmix[150], so genotypes and allelic imbalance were estimated from the same source. Second, BaseQTL for calling eQTLs has been designed to use hard genotype calls. To minimise the risk of false positive calls due to genotype miscalls, we only included candidate eQTLs with a genotype probability equal to or greater than 0.99. BaseQTL was then run using default parameters, except that we allowed for the minimum number of heterozygous individuals to be 3 (defaults 5) to run the model and the minimum number of heterozygous individuals with ASE information to be 3 (defaults to 5) to model ASE. Finally, we intersected

MatrixQTL or BaseQTL results with lead eQTLs from the multi-cohort study and determined Spearman's correlation of the effect sizes (Supplementary Fig. 2).

## CHi-C data processing for contact QTL detection

We applied post-hoc processing of our CHi-C data for contact QTL detection. First, we discarded all eQTLs that had a proxy SNP in tight LD that was proximal to the eGene promoter (<10 kb) in order to assay only truly distal effects, accounting for LD. In addition, we filtered out all eQTLs with potential effects on *DpnII* cut sites (either in the reference or alternative allele) that could cause allelic imbalance due to either (1) the technical ability of *DpnII* to fragment the library or (2) read mapping bias. In this filtering step, we also accounted for SNPs in LD that intersected the same *DpnII* fragment and combinations of SNPs within 4 bp that could potentially create a cut site (*GATC*). Finally, we estimated the affinity of our SureSelect capture probes for the *DpnII* CHi-C fragments containing either the reference or alternative haplotypes (all variants in $R^2 \geq 0.9$ with the lead eQTL) as the melting temperatures of the corresponding hybrids using the Tm_GC function from the R package TmCalculator v1.0.1, assuming Tris = 20 and Mg = 308. Haplotype-specific differences in the affinity were 0.18% on average and did not exceed 0.73% in all cases, and therefore we assumed them to have a negligible effect on capture efficiency. After filtering and processing, our total possible eQTL discovery set encompassed 1197 eQTLs and 12,295 proxy SNPs, either by direct capture or through capturing the eGene promoter (N = 1074 corresponding eGenes).

## BaseQTL detection of contact QTLs and ATAC QTLs

Prior to running BaseQTL on the CHi-C and ATAC-seq modalities, we first ran the refBias section of the BaseQTL input workflow using the scripts available from Gitlab (https://gitlab.com/evigorito/baseqtl_pipeline/-/tree/master/input/refbias). For each modality, we used the de-duplicated bam files and all genotyped variants passing the quality thresholds described above.

We processed the CHi-C reads such that we could detect (1) genetic effects on eQTL-eGene contacts and (2) genetic effects on non-eQTL-Gene contacts. For eQTL-eGene contacts, we processed the de-duplicated and filtered CHi-C bam files such that we only retained reads involving the *DpnII* fragment containing an eQTL (or a proxy SNP in tight LD, $r^2 \geq 0.9$) and a 5 kb region encompassing the eGene promoter. Where eGenes had more than one promoter, we compiled all reads across promoters. We did not require evidence of significant contact signals above the background. For non-eQTL variants, we looked for variants within other-end *DpnII* fragments of interactions involving captured gene promoters in our analysis (CHiCAGO score ≥ 5 in the consensus dataset). These variants were not associated with expression in the multi-cohort eQTL analysis[31,32], nor did they form significant interactions with eQTLs captured in our study design, thus there was no evidence for a role in regulating gene expression. To filter the bam files we first converted them to bedpe using the *bamtobed* function from BEDtools[141] v2.30.0. Then, we filtered the resultant bedpe against a reference bedpe with the paired regions that we wanted to test, using the *pairtopair* function from BEDtools. With the resultant read IDs we were then able to filter the original CHi-C bam files using SAMtools[137,141].

We extended BaseQTL to detect QTLs from CHi-C and ATAC reads. For these modalities, we model the uncertainty on the genotype calls obtained by imputation. BaseQTL combines total reads mapping interactions across individuals modelled by a negative binomial distribution $f_{NB}$ with reads mapping the alternative allele in heterozygous individuals modelled by a beta binomial distribution $f_{BB}$. The model estimates the logarithm of the allelic fold change. We define the allelic fold change $\beta_{aFC}$ as the ratio of the expected number of contacts in individuals homozygous for the alternative allele divided by the

expected number of contacts in individuals homozygous for the reference allele. In contrast to the original development case for BaseQTL[43], where we had to accommodate inferred haplotypes at the feature (expression across the whole gene body) as well as the regulatory SNP, which caused us to restrict to using genotypes called with high certainty, here the feature (contact or chromatin accessibility) is measured simply by read depth, which allowed us to make wider use of uncertain genotypes called through imputation. Using the same notation as Vigorito et al.[43], we adapted the likelihood as follows:

$$L(\beta_{aFC}, \gamma, \theta, \phi | c, n_1, G, X) \propto$$
$$\prod_{i=1}^{N} \left[ p(G_i = 1) f_{NB}(c_i | \beta_{aFC}, \phi, X) f_{BB}(n_{1i}; \pi, \theta, c_i) + \sum_{G_i = 0, 2} p(G_i = g) f_{NB}(c_i | \beta_{aFC}, \phi, X) \right];$$
(1)

$$\pi = \frac{exp(\alpha_{0i} + \beta_{aFC})}{1 + exp(\alpha_{0i} + \beta_{aFC})} \, for \, G_i \, heterozygous, \quad (2)$$

$$\pi = \frac{exp(\alpha_{0i})}{1 + exp(\alpha_{0i})} \, for \, G_i \, homozygous. \quad (3)$$

Where $\gamma$ is a vector of the regression parameters for covariates $\phi$ is the overdispersion parameter for the negative binomial distribution $\theta$ is the overdispersion parameter for the beta-binomial distribution $c_i$ is the total counts mapping the feature for individual $i$, $n_{1i}$ is the number of counts mapping the alternative allele of the candidate QTL in individual $i$, $X$ is a matrix of covariates $G_i$ is the genotype of the candidate QTL for individual i $p(G_i = g)$ is the genotype probability for each genotype class obtained from imputation $\alpha_{0i}$ is used to adjust for reference mapping bias, would be 0 in the absence of bias[43]. We used the cis-eQTL GTEx-derived default prior[151] on $\beta_{aFC}$ in BaseQTL, which is a mixture of two Gaussian distributions ($N(0, 0.001)$ and $N(0, 0.121)$), except that for this application, we considered a 50/50 mixture to reflect our expectation that 50% of the candidate QTLs have true effects. While the detection of contact QTLs was restricted to genetic variants intersecting the *DpnII* bait fragments included in the eQTL-CHi-C design, detection of ATAC QTLs included all genotyped genetic variants within 5 kb of ATAC-seq peaks (called using HMMRATAC[37] on the consensus ATAC-seq dataset across all 34 replicates). To obtain the number of reads mapping the alternative allele of the candidate QTL in heterozygous individuals, we used phASER[149] v1.1.1 from the filtered CHi-C bam files and ATAC-seq bam files, respectively. The same bam files were used to aggregate reads overlapping each feature. The code with the model for running BaseQTL on these modalities can be found at https://gitlab.com/evigorito/baseqtl_atac_chic_eqtl. Total read counts were adjusted by library size by providing it as a covariate. We did not correct for additional potential confounders. For all the modalities analysed by BaseQTL, significant associations were considered as those for which zero was excluded from the 99% credible interval for $\beta_{aFC}$. In addition, we discarded variants identified via an allelic imbalance in heterozygotes that had evidence of substantial baseline bias towards the reference genome (AI estimate < 0.4) and variants for which the model failed to converge (Rhat ≥ 1.01).

## Comparison of effects on promoter contact between eQTLs and non-eQTLs

We compared the BaseQTL results for eQTL-eGene contacts with non-eQTL-gene contacts in the following manner. First, we ensured that the non-eQTL variants (or their proxies in tight LD, $r^2 > 0.9$) were not associated with differential *DpnII* activity, as described in the CHi-C design above. Then, to rank candidate QTLs, we calculated the probability that the candidate SNP is a true eQTL. If we denote by $r_i \in \{0, 1\}$ the unknown truth for a SNP being (1) or not (0) a QTL, we can calculate $p_i = P(r_i = 1 | data)$ from our posterior samples by calculating the proportion of times that 0 was excluded from the 99% credible interval. To

**Table 1 | Cells, enzymes and genomic viewpoints used for 4C-seq**

| Locus | Type of experiment | Cell type | No. cells | No. reps | Viewpoint location in hg38 | Assessed cQTL and location in hg38 | Primer and location in hg38 | Restriction enzyme 1 | Restriction enzyme 2 |
|---|---|---|---|---|---|---|---|---|---|
| *THBS1* | Allele-specific | Primary monocytes | 5 M | 3 hets | chr15:39,315,236-39,315,577 | rs2033937 chr15:39,315,241 | TGAGTGATTGCAAATGGAAA chr15:39,315,242-39,315,261 | *DpnII* | *AseI* |
| *PCK2* | Allele-specific | Primary monocytes | 5 M | 3 hets | chr14:24,058,271-24,058,531 | rs7146599 chr14:24,058,463 | ACCAGACTTTCCACCAGAG chr14:24,058,443-24,058,462 | *DpnII* | *HpyCH4V* |
| *THBS1* | CRISPRi versus control cells | U-937 cells expressing dCas9 and sgRNA | 10 M | 3 test, 3 control | chr15:39,315,236-39,315,577 | NA | TGAGTGATTGCAAATGGAAA chr15:39,315,242-39,315,261 | *DpnII* | *AseI* |

do this with a manageable number of samples, we used a normal approximation to the marginal posterior distribution of the QTL effect. We denote this quantity as approx. pp, and we use it to rank SNPs. We computed $P(r_i = 0|data) = 1 - P(r_i = 1|data)$ and then used a permutation test to assess whether eQTL-eGene contacts have a lower mean of approx.pp compared with non-eQTL-gene contacts. The null distribution of mean approx.pp was obtained by generating 10,000 samples of 800 non-eQTL gene contacts each and taking the mean of the corresponding approx.pp for each sample. To control for read coverage that confers different statistical power of association detection, we sampled only those non-eQTL-gene contacts whose read counts mapped to the same quantile of the read count distribution of the corresponding eQTL-eGene contact.

## 4C-seq

4C-seq was performed as a validation of contact QTL effects on chromatin contact at rs2033937 (*THBS1*) and rs7146599 (*PCK2*) and to determine the effect of enhancer repression in the CRISPRi experiment (*THBS1* locus). For allele-specific 4C-seq, three individuals heterozygous for the variant in question were selected from the pool of monocyte donors. Around 5 million crosslinked nuclei, which were remaining after the CHi-C experiment, were used per experiment. For CRISPRi 4C-seq, we used 10 million U-937 cells expressing either (1) sgRNA targeting cQTLs in the *THBS1* locus or (2) sgRNA containing a scrambled, non-targeting sequence (see CRISPRi methods section below). For allele-specific identification of 4C-seq reads, the primary and secondary restriction enzyme (RE) and primer were selected such that the cQTL fell in the region between the primer and the primary restriction cut site in the viewpoint fragment (see Table 1 for details). This ensured that the cQTL always fell within the sequenced part of the viewpoint fragment.

4C-seq libraries were generated using a method based on published protocols[152,153]. Briefly, crosslinked cells were lysed in the same manner as for CHi-C and the chromatin was digested with the first restriction enzyme (200 U *DpnII*, for both loci) at 37 °C overnight. The enzyme was heat inactivated (65 °C for 20 min), and the nuclei were spun down and then re-suspended in a ligation mixture containing 50 units of DNA ligase (Invitrogen 15224017). After overnight incubation at 16 °C, crosslinks were reversed by the addition of Proteinase K (Sigma 3115879001) and incubation at 65 °C overnight. Samples were purified using SPRI beads (AMPure XP, Beckman Coulter), and up to 60 µg of the template was digested using 1U per µg template of the secondary restriction enzyme (AseI for *THBS1*, HpyCH4V for *PCK2*) at 37 °C overnight. The enzyme was heat-inactivated (65 °C for 20 min), and ligation was performed at a final DNA concentration of 5 ng/µL, using 2U per µg of T4 DNA ligase, overnight at 16 °C. Purification was performed the next day with Phenol Chloroform Isoamyl Alcohol (P:C:I) using MaXtract High-Density tubes (Qiagen). Final purification was performed using SPRI beads prior to library amplification, which followed the 2-step protocol described in Krijger et al.[153]. For the first PCR, four PCR reactions containing 200 ng of template were set up for each sample, and 16 cycles of PCR were performed using Expand Long

Template Polymerase Mix (Roche). The DNA was pooled and cleaned using 0.8X SPRI to remove long primers. For the second PCR (sequencing adaptor addition), up to 100 ng of template was used per reaction for 12 cycles of PCR. Final library cleanup was done using 0.8X SPRI, and the samples were sequenced using 150 bp paired-end reads on a NovaSeq platform (Novogene Europe).

Read 1 of the paired-end reads was used for all 4C-seq analyses. For allele-specific analyses, the raw reads were first split to each allele of the variant in question using *grep* from SeqKit[154], searching for the primer sequence and the intervening genomic sequence containing either allele of the variant, followed by the restriction cut site within the first 100 bp of the viewpoint fragment. 4C-seq reads were mapped and filtered using the pipe4C tool from the de Laat lab[153]. Following this, restriction digest files (rmap) were made for each combination of restriction enzymes using the Digester script from HiCUP[122] v0.9.2. These were used alongside the 4 C BAM files as input to the *coverageBed* function from BEDtools[141] v2.31.1 to count the number of reads in each BAM file corresponding to restriction fragments. The viewpoint fragment and neighbouring fragments were removed, with the remaining counts used as input to the 4Cker tool, a Hidden-Markov Model-based pipeline for identifying interactions in 4C-seq data[155]. Significant contacts within the vicinity of the viewpoint were detected using the *nearBaitAnalysis* function (1 Mb each side of the viewpoint) with option $k = 30$. To identify differential interactions, we used the *differentialAnalysis* function, modified to reflect the paired nature of our samples (*design = ~subject + condition*). The right-left ratio in the *THBS1* locus was calculated from the 4Cker *nearBaitAnalysis* result, dividing the sum of all counts 1 Mb to the right of the viewpoint by the sum of all fragments 1 Mb to the left of the viewpoint per sample. For plotting purposes, we converted read counts to *rlog* using the DESeq2[148] R package v1.42.0.

## Detection of trimodal QTLs by the GUESS framework

To generate the regions for GUESS, windows were defined at regions in close proximity to eQTL-containing enhancers targeted in the CHi-C design and peaks of open chromatin in the ATAC-seq data. Specifically, we expanded 5 kb on either side from the CHi-C baited *DpnII* fragments and from the ATAC-seq peaks and took the minimum intersection of these two ranges. We considered all genotyped variants within these windows for the GUESS analysis. To avoid the impact of *DpnII* cut site perturbation in the CHi-C data, we removed windows where any of the considered variants affected or generated a *DpnII* cut site at the CHi-C bait fragment. We additionally did not consider windows where any of the variants fell within 10 kb of a promoter for the eGene.

To control for mapping bias to the reference genome, the raw sequencing data for CHi-C, ATAC-seq and RNA-seq were first processed using the WASP pipeline[156]. We then performed variance stabilisation and normalisation on the total counts for CHi-C, ATAC-seq and RNA-seq using the *rlog* approach implemented in the R package DESeq2[148]. For each window, we then ran GUESS[45,46] for 20,000 iterations, of which 5000 were burn-in with three parallel chains to avoid the algorithm being stuck in a local mode. A priori, and independently

of the number of genetic variants included in each window, the expected number of trimodal QTLs was set equal to 1 as well as its standard deviation, which implies a range between zero and four for the number of significant trimodal QTLs. Prior to the analysis, the genetic variants were standardised to place appropriate prior mass on reasonable values of the non-zero regression coefficients. Moreover, we subtracted the mean of the rlog transformed total counts, implicitly specifying a non-informative prior on the intercept for each molecular trait. We did not perform correction for known confounders since our donors were age-group, sex and population-matched and did not correct for latent factors given the lack of suitable methodology for the joint correction of multiple modalities. We also did not filter genetic variants by LD since the GUESS algorithm is robust to extreme multi-collinearity given that marginal likelihood is calculated using the technique of QR matrix decomposition and parallel tempering allows the inclusion of collinear predictors in different chains but not on the same one. Swapping selected genetic variants between chains allows the algorithm to explore the space of collinear predictors without incurring rank deficiency. For each window analysed and the chain with temperature equal to one, we recorded the best model visited, i.e., the combination of genetic variants visited by the algorithm with the largest marginal likelihood score, and the marginal posterior probability of inclusion, i.e., the marginal association strength of each genetic variant with all molecular traits. If the best model visited was the null model, we declared no associations. Otherwise, we calculated the FDR threshold for the marginal posterior probability of inclusion as:

$$\text{argmax}_j \sum_j \left(1 - mPPI_{(j)}\right) \leq q, \qquad (4)$$

where $mPPI_{(j)}$ is the $j$th-ordered marginal posterior probability of inclusion sorted in non-decreasing order and $q$ is the designed level of False Discovery Rate, set at 0.05[47]. While the best model visited can overlap with the model obtained by filtering the marginal posterior probability of inclusion by the specified FDR level, in general, they do not coincide. We also generated the posterior effects size and its standard error averaging over all models visited in a post-processing step, thus performing Bayesian Model Averaging[157]. Finally, in analogy with classical statistics, we defined the posterior expectation of the $z$-score as:

$$\text{E}(z_{jk}|Y) \approx \frac{\text{E}(\beta_{jk}|Y)}{\left\{\text{E}(\sigma^2(\beta_{jk})|Y)\right\}^{1/2}}, \qquad (5)$$

where $\beta_{jk}$ is the effect size of the $j$th genetic variant on the $k$th molecular trait in each window and $\sigma^2(\beta_{jk})$ is the variance of the effect size for the $j$th genetic variant on the kth molecular trait.

We further filtered the GUESS results after FDR selection as follows: firstly, we discarded variants that had evidence of baseline allelic imbalance at the feature in question (AI estimate < 0.4), as determined by the refBias pipeline of BaseQTL. Finally, we filtered out QTLs with very small absolute effect sizes to ensure the replicability of the results with BaseQTL. For this, we examined the external concordance of variant effects on CHi-C, ATAC and RNA reads between GUESS and BaseQTL (illustrated in Supplementary Fig. 5). We tuned the cutoff of the beta value to maximise this concordance without losing sensitivity, resulting in the following beta cutoffs on significant trimodal QTLs: CHiC beta ≥ 0.006, ATAC beta ≥ 0.001 and RNA beta ≥ 0.004.

To validate a subset of GUESS results with a different statistical strategy, we performed multivariate analysis of variance (MANOVA)[83] on the lead trimodal QTL identified by GUESS in each window, treating the other trimodal QTLs in the same window (if detected at 5% FDR) as covariates (MANCOVA). We also assessed how many of the lead

trimodal QTLs could be validated using a univariate approach (ANOVA/ANCOVA) (Supplementary Fig. 5C, D).

## Causal mediation analysis

We used the R package mediation to perform causal mediation analysis in the potential outcome framework[158]. The causal mediation analysis was set up as follows. Let $M_i(x)$ represent the potential mediator value for individual donor $i$ if the individual's treatment status is $X_i = x$. Let $Y_i(x)$ denote the potential outcome value for individual donor $i$ if $X_i = x$ and individual $i$ has a mediator value $M_i = m$. For a binary treatment, the causal mediation effect for individual donor $i$ is captured by the difference between the observed outcome and the counterfactual outcome as if the individual's treatment status remains the same, but the mediator value equals the value under the other treatment status:

$$\delta_i(t) = Y_i(x, M_i(1)) - Y_i(x, M_i(0)), \text{x} = \{0,1\}. \qquad (6)$$

If $x = 0$, $Y_i(0, M_i(0))$ is an observed outcome and $Y_i(0, M_i(1))$ is a counterfactual outcome, and vice versa if $x = 1$. The *average causal treatment effect* (ACME) for treatment t is defined as $\delta(x) = E\{\delta_i(x)\}$ and is computed separately for treatment (ACME(1)) and control conditions (ACME(0)). The *average direct effect* (ADE) is defined as $\zeta(x) = E\{Y_i(1, M_i(x)) - Y_i(0, M_i(x))\}$, $\text{x} = \{0,1\}$. The total causal effect (TE) of the treatment is defined as $\tau = \delta(x) + \zeta(1 - x)$.

To compute the ACME, two regression equations are first specified and fitted, one for the mediator and one for the outcome. Specifically:

$$M_i = \alpha_0 + \alpha_1 X_i + \alpha_2 C_i + \varepsilon_{im}, \qquad (7)$$

$$Y_i = \beta_0 + \beta_1 M_i + \beta_2 X_i + \beta_3 C_i + \varepsilon_{iy}, \qquad (8)$$

where $C_i$ is a vector of observed confounders for individual $i$ and $\varepsilon_{im}$ and $\varepsilon_{iy}$ are independent error terms. Since in our analysis, the mediators and the outcomes are continuous, the ACME estimand is obtained as the product of the effects:

$$ACME(1) = \alpha_1\beta_1, \qquad (9)$$

$$ACME(0) = \alpha_1\beta_2. \qquad (10)$$

In model $M_{ATAC} \to CHi\text{-}C$, the mediator and outcome equations are:

$$ATAC_i = \alpha_0 + \alpha_1 PC_{i,j} + \varepsilon_{i,ATAC}, \qquad (11)$$

$$CHiC_i = \beta_0 + \beta_1 ATAC_i + \beta_2 PC_{i,j} + \varepsilon_{i,CHiC}, \qquad (12)$$

where $PC_{i,j}$ is the $j$th principal component for individual $i$.

The models $M_{ATAC} \to RNA$ and $M_{CHi\text{-}C} \to RNA$ are analogues to the $M_{ATAC} \to CHi\text{-}C$.

For the model $M_{CHi\text{-}C(ATAC)} \to RNA_{(ATAC)}$, where ATAC is an observed confounder of both chromatin interactions and gene expression, the two equations are:

$$CHiC_i = \alpha_0 + \alpha_1 PC_{i,j} + \alpha_2 ATAC_i + \varepsilon_{i,CHiC}, \qquad (13)$$

$$RNA_i = \beta_0 + \beta_1 CHiC_i + \beta_2 PC_{i,j} + \beta_3 ATAC_i + \varepsilon_{i,RNA}. \qquad (14)$$

Since in our models, the treatment is continuous (PCs), we calculate ACME(PC$_j$), $j = 1,...,J$, where $J$ is the number of PCs that retain a cumulative proportion of variance explained ≤ 0.99. We tested the models separately on each PC and selected models with a $p$ value of the average causal mediation effect (pACME-value) <0.05. Confidence intervals of ACME(PC$_j$), ADE(PC$_j$) and TE(PC$_j$) and corresponding $p$ values are obtained by specifying the argument *boot = TRUE* for the non-parametric bootstrap confidence intervals, *boot.ci.type = BCA* for the bias-corrected and accelerated (BCa) confidence intervals. Finally, we set *sims = 1000* for the number of bootstrap samples. We checked if the Total Effect has a significant $p$ value (TE $p$ value ≤ 0.05) and whether the $p$ value of the Average Direct Effect (ADE $p$ value) is significant at the same level. If TE $p$ value ≤ 0.05 and ACME $p$ value ≤ 0.05, we declared a significant Causal Mediation Effect. By looking at ADE $p$ value, we also distinguished between a fully mediated model (ADE $p$ value > 0.05) and a partially mediated one (ADE $p$ value ≤ 0.05).

Causal mediation relies on stronger assumptions than traditional causal inference and, in particular, the assumption of no unmeasured confounding known as 'sequential ignorability'[159,160]. In practice, A1) there shouldn't be any confounder (U1) between the outcome and the treatment, A2) between the mediator and the treatment (U3) and A3) between the outcome and the mediator (U2). In our set-up, since the genotypes are assigned at conception (like in an RCT with random treatment allocation), there is less risk of U1 and U3, but U2 is hard to control. We use the function medsens() in the *mediation* R package to assess the sensitivity of the mediation model to assumption A3. The sensitivity analysis relies on the idea that if there is a confounding effect on both the outcome and the mediator, the correlation between $\varepsilon_m$ and $\varepsilon_y$ is large, and knowing this level of correlation, the mediation effect can be calculated without bias. In practice, *medsens()* returns the level of correlation that is required to change the sign of the estimated level of mediation. The larger this level of correlation, the less sensitive the estimated level of mediation to unmeasured confounders U2. We report this value in Supplementary Data 11, where sensitivity values ≥ 0.3 in absolute value are regarded as robust to unmeasured confounders U2. For instance, for the trimodal *THBS1* locus with a significant full mediation model MI (M$_{ATAC}$→CHi-C), the sensitivity of ACME is 0.4 (Supplementary Fig. 6B), while for the trimodal *NFE2L3* locus with the model MII (M$_{ATAC}$ → RNA) the sensitivity of ACME is 0.3 (Supplementary Fig. 6D). For the trimodal locus *SPSB1* with the significant mediation model M$_{CHi-C}$ → RNA, the sensitivity of ACME is 0.4 (Supplementary Fig. 6E). For the *ABHD2* locus with a significant full mediation model MI (M$_{ATAC}$→CHi-C), the sensitivity of ACME is 0.3 (Supplementary Fig. 6F).

### CRISPR interference in the *THBS1* locus

U-937 cells were cultured in RPMI media, supplemented with 10% fetal bovine serum (FBS) and 1% penicillin-streptomycin. HEK293 cells were cultured in DMEM, supplemented with 10% FBS and 1% penicillin-streptomycin. The cells were maintained at 37 °C in a humidified incubator with 5% CO$_2$ and were passaged at 80% confluence. Cells were regularly checked for mycoplasma infection.

To generate cell lines expressing dCas9 KRAB, the lentiviral puromycin resistance vector pLV-dCas9-KRAB (Addgene plasmid #99372) was used. For sgRNA expression, we further used the lentiviral hygromycin resistance vector pLV-sgRNA (Addgene plasmid #62205). We cloned the following sgRNA targeting cQTLs in the *THBS1* locus separately into plasmid #62205: TATGCTCTTCAGAACAAACC (targeting chr15:39607469-39607488); AGCTGATGAATGCCCATACT (targeting chr15:39607555-39607574) and AATTTAGGCCTCCTAATATG (targeting chr15:39602387-39602406). We combined the three plasmids containing each sgRNA in equimolar quantities, making a pool of cQTL sgRNA plasmids. As a control, we cloned a sgRNA with a scrambled, non-targeting sequence into plasmid #62205 (CTTAGTTACGCGTGGACGA).

Lentivirus was produced for the pLV-dCas9-KRAB plasmid (#99372) and the cloned sgRNA plasmids (#62205) using a third-generation system. HEK293 cells were seeded to 10 million cells per vector in 15 cm round plates in a 20 mL volume of complete media and incubated overnight. The media was replaced with 20 mL fresh media with FBS but without penicillin-streptomycin. Then, 6 μg of plasmid DNA was combined with 1.5 μg of REV packaging plasmid (Addgene # 12253), 3 μg pRRE packaging plasmid (Addgene # 12253) and 1.5 μg VSV-G envelope (Addgene # 12259) in 2 mL serum-free DMEM without phenol red in a sterile 15 mL tube. After filter sterilising, we added 72 μL of 1 mg/ml Polyethyleneimine linear MW 25 K (PEI; Fisher Scientific 11460630) to the tube, vortexed and left at room temperature for 10–15 min. The DNA/PEI mix was added to the cells in a dropwise fashion using a Pasteur pipette. The cells were incubated overnight before a media change. After a 48-h incubation, the media containing lentivirus was harvested and filtered (0.45 μm filter) before use.

The first round of transduction generated U-937 cells expressing dCas9-KRAB. Before transduction, 300,000 U-937 cells were seeded into 3 mL media in a six-well plate. The next day, cells were spun down and resuspended in 2 mL of media with FBS but no penicillin-streptomycin. After 2 h, 24 μL of polybrene was added (Hexadimethrine bromide, Sigma H9268-5G, prepared at 1 mg/mL in 0.9% NaCl), followed by 1 mL of lentivirus. Cells were incubated overnight before changing the media (RPMI media + 10% FBS and 1% penicillin-streptomycin). After a further 48 h, the cells underwent selection with puromycin (250 μg/mL) for several days until cells that had not integrated the plasmid had died and transduced cell viability returned to high levels. Subsequently, the cells were transduced with the sgRNA lentivirus (either the pool of cQTL-targeting sgRNA or the scrambled control sgRNA) in the same manner as for dCas9 KRAB. The cells were then selected with hygromycin (1 μg/mL) until viability returned to high levels. Three independent rounds of sgRNA transductions were performed to generate biological triplicates.

Total RNA was extracted from the transduced U-937 cells using the RNeasy Mini Kit (Qiagen). cDNA synthesis was performed with the High-Capacity cDNA Reverse Transcription Kit (Applied Biosystems). Quantitative PCR was conducted using SYBR Green Master Mix (Thermo Fisher Scientific) on a QuantStudio 6 Flex Real-Time PCR System (Applied Biosystems). *THBS1* expression levels were normalised to the geometric mean of three housekeeping genes: *ATP*, *TOP1* and *GAPDH*. Relative expression levels were calculated using the ΔΔCt method.

For 4C-seq, 10 million transduced U-937 cells were fixed with 1% formaldehyde. Chromatin was processed as previously described (see the 4C-seq section above), with the viewpoint centred on rs2033937. For ATAC-seq, transduced cells were processed according to the protocol described above (see the ATAC-seq section). Both 4C-seq and ATAC-seq experiments were also performed in biological triplicate.

### CTCF chromatin immunoprecipitation-sequencing (ChIP-seq)

CTCF ChIP-seq was performed on crosslinked material from three of the genotyped primary human monocyte samples also used for CHi-C, ATAC-seq and RNA-seq, following the Chipmentation methodology (SOP version 1.14) of the Bock lab[161]. These samples were selected to be heterozygous for the cQTL rs7146599 (*PCK2* locus). Briefly, we sonicated the chromatin from 2 million cells for each sample in 400 μL sonication buffer using 10 cycles of 30 s on and 30 s off on the Diagenode Bioruptor Pico. We then performed immunoprecipitation (IP) on 100 μL of sonicated material, corresponding to around 500 K cells, overnight with either 1 μL of non-specific IgG antibody (Fisher Scientific 10610274) or 4 μL CTCF antibody (Merk 07-729) and 25 μL of blocked protein A beads (Fisher Scientific 10334693). We retained 5 μL of sonicated material as an input control sample. The next day, the IP samples were washed using a series of buffers according to the SOP (RIPA-LS, RIPA-HS, RIPA-LiCl and 10 mM Tris-HCl, pH 8). Tagmentation

was done on the IP samples using 1 µL of homemade Tn5 for 10 min at 37 °C, followed by further washes (RIPA-LS and TE buffer), according to the protocol. For the input samples, 2.5 ng of material was tagmented in a total 5 µL reaction using 1 µL of 1/10 Tn5 for 5 min at 55 °C. De-crosslinking was performed for all IP and input samples overnight in 48 µL ChIP elution buffer and 2 µL proteinase K (Sigma 3115879001) at 65 °C. The next day, the DNA was cleaned using 1X SPRI beads (AMPure XP, Beckman Coulter). For sequencing, libraries were amplified using 12 cycles of PCR and cleaned using 1.1X SPRI beads, followed by 150 bp paired-end sequencing on a NovaSeq platform (Novogene Europe).

We used the ChIP-AP pipeline[162] v5.4 for fully integrated ChIP-seq QC and peak calling, using the input chromatin samples as controls. To count the number of allele-specific reads at rs7146599, the trimmed but non-deduplicated reads were processed using the WASP pipeline to remove mapping biases[156], with final de-duplication performed after removing the reads that mapped in a different location upon allele flipping. PhASER[149] was used to count allele-specific reads at heterozygous variants. We used a Python script to split bam files based on haplotype (https://github.com/luntergroup/bamsplit), followed by merging across samples with SAMtools[131,137] v1.9 and generating bigwig files using *bamCoverage* from deepTools[133] v3.3.1 to enable visualisation of CTCF binding at rs7146599 in the *PCK2* locus.

## Downstream computational analysis of cQTLs

For the following analyses, we combined the contact eQTLs from BaseQTL and the trimodal QTLs from GUESS, making a total of 641 cQTLs.

## Identification of TF binding at cQTLs via ChIP-seq peaks and ATAC-seq binding predictions

Non-redundant peaks for all cell types and factors in the ReMap 2022 catalogue[48] were downloaded in GRCh38 from https://remap.univ-amu.fr/download_page. We supplemented the ReMap dataset with CTCF ChIP-seq peaks generated on three primary monocyte samples from the present study (see ChIP-seq methods above). To build a dataset of experimentally determined TF binding in monocytes, we filtered the ReMap ChIP-seq dataset, with the inclusion of our CTCF ChIP-seq data, to the following monocyte cell types: 'monocyte', 'CD14', 'THP-1', 'U-937' and 'HL-60'. We then merged all peaks per TF using *merge* from BEDtools[141] v2.30.0.

We used two methodologies to predict TF binding from the ATAC-seq data generated in the present study: TOBIAS[49] and MaxATAC[50]. We ran the TOBIAS Snakemake pipeline (available at https://github.com/loosolab/TOBIAS_snakemake) on cleaned ATAC-seq bam files for all 34 ATAC-seq replicates and detected the binding of all TFs with motifs in the Jaspar 2020 dataset[163] at ATAC-seq footprints. For MaxATAC, we first generated bigwig files per replicate using the *prepare* function with clean ATAC-seq bam files as input. Then, we used *average* to make an averaged bigwig across all replicates, followed by *predict*, against all 127 pre-trained TF models available from the MaxATAC authors. We then combined the predicted 'bound' TF locations with the TOBIAS footprints using *merge* from BEDtools[141] v2.30.0 to generate a final dataset of 'ATAC-seq predicted' TF binding sites.

We combined the ChIP-seq peaks, and ATAC-seq predicted peaks into one table but kept the two sources separate (i.e. specifying if the peak set came from ChIP-seq or ATAC-seq). We then determined monocyte TF enrichment at 256 bp windows around cQTLs (QTL ± 128 bp) using the Remapenrich R package (https://github.com/remap-cisreg/ReMapEnrich) v0.99.0. We used a background Universe consisting of all the variant locations submitted to the GUESS and BaseQTL analyses in GRCh38 (filtered for *DpnII* effects and distal from eGene promoter; also ±128 bp), with 100 shuffles. We considered significant enrichment at $-\log_{10}$ adjusted *p* value (Q significance) > 5 and the number of overlapping

QTLs ≥ 5. Intersections between cQTLs (±128 bp) and TF binding sites were further explored for evidence of TF perturbation by the cQTL (see the following section).

To determine TF enrichment at cQTLs across all cell types, we used the whole Remap 2022 catalogue[48] (1171 TFs and 726 cell types) without first selecting cell types of interest. The ReMapEnrich R package (https://github.com/remap-cisreg/ReMapEnrich) v0.99.0 was used to determine enrichment using the same settings as for the monocyte dataset.

## Prediction of TF binding perturbation at QTLs by deep learning tools

To obtain allele-specific predictions of TF binding from the Enformer[52], we used the publicly available precomputed Enformer results for all SNPs with MAF > 0.5% (https://console.cloud.google.com/storage/browser/dm-enformer/variant-scores/1000-genomes/enformer). DeepSea[51] predictions were queried using the online interface (https://hb.flatironinstitute.org/deepsea/). We restricted the analysis to the 179 TFs in the Enformer training set and 77 TFs in the DeepSea training set, for which we could obtain independent evidence of binding to the cQTLs and ATAC-QTLs of interest from at least one of the following sources: (1) ChIP-seq data in monocytes, (2) predictions from ATAC-seq footprint analysis by TOBIAS[49], (3) MaxATAC[50] predictions, as described above.

We followed ref. 51 to express the predictions of variant effect on TF binding predicted by each model in terms of the TF binding perturbation score, defined as:

$$\text{Perturbation Score}(model, TF) = \max\left(\text{SAD}_{model, TF, i} * \text{SAR}_{model, TF, i}\right)$$
(15)

across all ChIP datasets *i* for a given TF available in the training set for either model (DeepSea or Enformer), and for each dataset:

$$\text{SAD}_{model, i, TF} = \text{alt allele score}_{model, i, TF} - \text{ref allele score}_{model, i, TF},$$
(16)

$$\text{SAR}_{model, i, TF} = \log_2\left(\text{alt allele score}_{model, i, TF} + 1\right) - \log_2\left(\text{ref allele score}_{model, i, TF} + 1\right),$$
(17)

where SAD and SAR are SNP activity difference and ratio, respectively.

To estimate the background range for each TF, we generated Enformer and DeepSea TF perturbation scores for a random set of ~275,000 SNPs sampled from across the genome. Variants whose perturbation scores for a given TF exceeded the top 1% percentile of the background range for both DeepSea and Enformer were considered TF-perturbing.

## Disruption of TF binding motifs by cQTLs

TF binding motif disruption by cQTLs was detected using the motif-breakR package[164] in R using the motif data sources 'HOCOMOCOv11-core-A', 'HOCOMOCOv11-core-B', 'HOCOMOCOv11-core-C' from the R package MotifDb. Motif matches were detected at a *p* value threshold of 1e$^{-4}$ using the 'ic' method against a flat genomic background. The predicted 'strong' variant effects on the TF binding motif (corresponding to the absolute difference in position weight matrix match scores of at least 0.7) were used in downstream analysis.

## GWAS integration

For each cQTL, we identified proxy variants at unphased $r^2 \geq 0.8$ using PLINK v2[118] with a 1000 Genomes Phase 3 reference panel[119]. We then obtained a direct overlap of cQTLs and their proxy variants with variants listed in the GWAS Catalog[165] (downloaded on 16th June 2023).

Since adapting GUESS association statistics for formal GWAS colocalisation analysis is non-trivial, we instead sought evidence of colocalisation between GWAS and public eQTL data in monocytes or whole blood that implicated our cQTL variants as eQTLs for the same eGenes (credible interval > 95%). This analysis required GWAS summary statistics, which were publicly available for several white blood cell count GWAS, but not the other traits used as examples in our study. We queried precomputed coloc[166] results for these datasets, eGenes and variants using the Open Targets Genetics platform[167].

### Reporting summary

Further information on research design is available in the Nature Portfolio Reporting Summary linked to this article.

## Data availability

Source data for figure panels and large Supplementary Data (such as eQTL CHi-C design, CHi-C significant interactions, ATAC-seq peaks and TF footprinting) are available on the Open Science Framework (OSF) at https://osf.io/szntj/ (https://doi.org/10.17605/OSF.IO/SZNTJ)[167]. Raw sequencing data from human monocytes generated in this study are available in the European Genome-Phenome Archive (EGA) under managed access, in accordance with the donor terms of consent. The data can be found under Dataset ID EGAD50000001116. To request access to these data, please contact the Data Access Committee at eQTL-CHiC-DAC-WC@groups.imperial.ac.uk. An application form will be provided within seven working days of the request. Submitted forms signed by the requestor(s) and an authorised institutional representative will be reviewed within one calendar month, and access to data download through EGA will be granted, provided all access conditions are met. 4C-seq and ATAC-seq data from U-937 cells are deposited in GEO for unrestricted access, accession numbers GSE281908 and GSE281909. Source data are provided with this paper.

## Code availability

The code used to generate the results in this study is available at the following repositories: https://github.com/FunctionalGeneControl/contactQTLs (eQTL CHi-C design and data processing; stable release is available on Zenodo (https://doi.org/10.5281/zenodo.14446477)[168]), https://github.com/pavarte/PCHIC-ABC-Prediction (CHi-C ABC analysis), https://github.com/lb664/R2GUESS (GUESS framework), and https://gitlab.com/evigorito/baseqtl_atac_chic_eqtl (BaseQTL extended to CHi-C and ATAC-seq modalities).

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

## Acknowledgements

The authors would like to thank Oliver Stegle, Laurence Game and Ivan Andrew for advice and technical assistance and Matthias Merkenschlager for the critical reading of the manuscript. We thank the UK's National Institute for Health and Care Research (NIHR) BioResource volunteers for their participation and gratefully acknowledge NIHR BioResource centres, UK's National Health Service (NHS) Trusts and staff for their contribution. We thank the NIHR, NHS Blood and Transplant, and Health Data Research UK as part of the Digital Innovation Hub Programme. The views expressed are those of the authors and not necessarily those of the NHS, the NIHR or the Department of Health and Social Care. This work was supported by the Medical Research Council (MRC) of the UK through MRC Investigator funding to M.S. (MC-A652-5QA20). L.B. gratefully acknowledges funding from the MRC (MR/W029790/1), the Alan Turing Institute under the Engineering and Physical Sciences Research Council (EPSRC) grant EP/N510129/1, the Marmaduke Sheild Fund and the NIHR Cambridge Biomedical Research Centre (NIHR203312). M.F. is supported by the British Heart Foundation (BHF; FS/18/53/33863), the BHF Cambridge Centre for Research Excellence (RE/18/1/34212) and the NIHR Exeter Biomedical Research Centre. C.W. and E.V. are supported by the MRC (MC_UU_00002/4) and C.W. by the Wellcome Trust (WT220788).

## Author contributions

Design and conceptualisation: H.R.-J., M.S., L.B., and V.M. Wet-lab experiments: H.R.-J., L.T.C., F.B., E.W., Z.B., V.M., and V.X.H.T. Computational analysis: H.R.-J., L.B., C.K.S., E.V., M.S., V.M., A.H., P.A., L.R., and A.L. Experimental and computational methodology: H.R.-J., L.B., V.M., E.V., C.W., P.A., and M.D.R. Resources: F.B., M.F., and R.K. Supervision: M.S., L.B., V.M., E.V., H.R.-J., and C.W. Writing: H.R.-J., M.S., V.M., L.B., and E.V., with comments from all authors.

## Competing interests

M.S. is a shareholder of Enhanc3D Genomics Ltd. C.W. is supported by GSK and MSD and is a part-time employee of GSK. GSK had no role in this study or the decision to publish. The remaining authors declare no competing interests.

## Additional information

[1]MRC Laboratory of Medical Sciences, London, UK. [2]Institute of Clinical Sciences, Imperial College Faculty of Medicine, London, UK. [3]Computational Neurobiology, VIB Center for Molecular Neurology, VIB, Antwerp, Belgium. [4]Computational Neurobiology, Department of Biomedical Sciences, University of Antwerp, Antwerp, Belgium. [5]Department of Brain Sciences, Faculty of Medicine, Imperial College London, London, UK. [6]Department of Haematology, University of Cambridge, Cambridge Biomedical Campus, Cambridge, UK. [7]National Health Service (NHS) Blood and Transplant, Cambridge Biomedical Campus, Cambridge, UK. [8]EMBL-EBI, Wellcome Genome Campus, Cambridge, UK. [9]Department of Clinical and Biomedical Sciences, Faculty of Health and Life Sciences, University of Exeter Medical School, Exeter, UK. [10]Cambridge Institute of Therapeutic Immunology & Infectious Disease (CITIID), Jeffrey Cheah Biomedical Centre, University of Cambridge, Cambridge, UK. [11]MRC Biostatistics Unit, School of Clinical Medicine, University of Cambridge, Cambridge, UK. [12]Department of Medical Genetics, School of Clinical Medicine, University of Cambridge, Cambridge, UK. [13]The Alan Turing Institute, London, UK. [14]Present address: Department of Internal Medicine, Erasmus MC, Rotterdam, The Netherlands. [15]Present address: LKS Faculty of Medicine, the University of Hong Kong, Hong Kong, Hong Kong. [16]Present address: Cyted, Cambridge, UK. [17]Present address: University of Kent, Canterbury, UK. [18]Present address: Swiss Federal Administration, Bern, Switzerland. [19]Present address: Institute of Computational Biology, Helmholtz Zentrum München and Ludwig Maximilians University Munich, Faculty of Medicine, Munich, Germany. [20]Present address: University Hospital Antwerp (UZA), Antwerp, Belgium. [21]Present address: University of Antwerp, Antwerp, Belgium. [22]Present address: Hummingbird Bioscience, Singapore, Singapore. [23]These authors contributed equally: Valeriya Malysheva, Leonardo Bottolo, Elena Vigorito. [24]These authors jointly supervised this work: Valeriya Malysheva, Leonardo Bottolo, Elena Vigorito, Mikhail Spivakov. ✉e-mail: h.ray-jones@erasmusmc.nl; lb664@cam.ac.uk; mikhail.spivakov@lms.mrc.ac.uk

