## [Peer Review file · Nature Communications]

Genetic coupling of enhancer activity and connectivity in gene expression control

Corresponding Author: Dr Mikhail Spivakov

Version 0:

Reviewer comments:

Reviewer #1

(Remarks to the Author)

The authors have addressed all of my concerns.

I have one minor point of clarity that I would like them to make and I don't need to review it.

Response 8 "The effect of covariates is cancelled when modelling within-individual variation, which results in substantial statistical power gain."

This is true for the allelic component but there are also covariates for the across individual component. The equation itself states " γ is a vector of the regression parameters for covariates". Were any used in BaseQTL? Or is it similar to their application of GUESS where the authors have opted to not include any covariates "We did not perform correction for known confounders ... and didn't correct for latent factors".

Reviewer #3

(Remarks to the Author)

First of all I would like to thank the authors for thoroughly revising their manuscript based on the comments given by all reviewers. It is clear the authors took the time to do a thorough rebuttal of the previous manuscript.

The manuscript has definitely improved and the experimental validations as well as extra analyses clearly strengthen the manuscript.

In my opinion, the authors addressed most of my other concerns. The only comment that the authors could not address, is the comment concerning the modest cohort size. However, as mentioned by themselves, this is a comment they cannot easily tackle. Therefore it was crucial for them to answer all the other questions/comments raised (eg. experimental validation of the PCK2 hypothesis, revised TF analysis), which they did.

Minor comments

Typo in caption of Supplemental Figure 2

- Chicaco instead of ChiCAGO and non-working references (33, 34)

Reviewer #4

(Remarks to the Author)

Review of NCOMMS-24-52355-T, Ray-Jones et al.

Since the original reviewer #2 was not available to review the revised manuscript, I am specifically reviewing the authors' response to the original comments from reviewer #2.

The authors have substantially edited the manuscript in response to both reviewers' comments. New analyses have been added, old ones revisited and extended/revised, and key points have been explained and clarified throughout the manuscript. The authors have also performed new experiments to follow-up on some of the original results, including a CRISPRi analysis at a candidate locus to demonstrate a causal link between enhancer activity and enhancer-promoter looping, which have now been included. Overall, the authors have responded to all comments carefully and at length, illustrating also the broader context in which their results fall, by referencing many other published works throughout their response. I feel their responses to the comments are adequate and their choices and approaches well justified.

Both reviewers raised concerns about the incremental nature of the work, i.e. novelty. Similarly, both highlighted the sample size (N=34) as a potential issue. In my view the authors make a reasonable case why it is not feasible for them to increase the sample size for this manuscript. While N=34 is certainly on the small side for a molecular QTL study, increasing it to e.g. 60 (which is also small) likely wouldn't drastically change the conclusions of the manuscript, even if more QTLs would be detected. The study thus serves as a proof-of-concept to demonstrate that shared genetic effects on enhancer-promoter contacts and enhancer activation exists.

We have addressed Reviewer 1's comment as follows:

Response 8 "The effect of covariates is cancelled when modelling within-individual variation, which results in substantial statistical power gain."

This is true for the allelic component but there are also covariates for the across individual component. The equation itself states " γ is a vector of the regression parameters for covariates". Were any used in BaseQTL? Or is it similar to their application of GUESS where the authors have opted to not include any covariates "We did not perform correction for known confounders ... and didn't correct for latent factors".

We have now explicitly clarified that adjustment for library size in BaseQTL was made by including it as a covariate, and no other covariates were included (p.27).